# Kinetic Langevin Diffusion for Crystalline Materials Generation

**François Cornet** [* 1]  **Federico Bergamin** [* 1 2]
**Arghya Bhowmik** [1]  **Juan Maria Garcia Lastra** [1]  **Jes Frellsen** [1 2]  **Mikkel N. Schmidt** [1 2]

## Abstract

Generative modeling of crystalline materials using diffusion models presents a series of challenges: the data distribution is characterized by inherent symmetries and involves multiple modalities, with some defined on specific manifolds. Notably, the treatment of fractional coordinates representing atomic positions in the unit cell requires careful consideration, as they lie on a hypertorus. In this work, we introduce Kinetic Langevin Diffusion for Materials (KLDM), a novel diffusion model for crystalline materials generation, where the key innovation resides in the modeling of the coordinates. Instead of resorting to Riemannian diffusion on the hypertorus directly, we generalize Trivialized Diffusion Model (TDM) to account for the symmetries inherent to crystals. By coupling coordinates with auxiliary Euclidean variables representing velocities, the diffusion process is now offset to a flat space. This allows us to effectively perform diffusion on the hypertorus while providing a training objective that accounts for the periodic translation symmetry of the true data distribution. We evaluate KLDM on both Crystal Structure Prediction (CSP) and De-novo Generation (DNG) tasks, demonstrating its competitive performance with current state-of-the-art models.

## 1. Introduction

The discovery of novel compounds with desired properties is critical to several scientific fields, such as molecular discovery (Bilodeau et al., 2022) and materials design (Merchant et al., 2023; Zeni et al., 2025). In the case of crystalline materials, the search space is vast, but only a fraction of it is physically plausible. The main challenges are to efficiently search the space for *feasible* materials and to accurately estimate their properties. Conventional approaches usually combine random structure search methods with *ab-initio* Quantum Mechanics (QM) methods (Oganov et al., 2019), such as Density Functional Theory (DFT) (Kohn & Sham, 1965); however, structural optimization and property evaluation with DFT can be computationally expensive. Recently, the field has witnessed a paradigm shift where deep generative models have been introduced to supplement traditional search methods (Anstine & Isayev, 2023), such as random search (Pickard & Needs, 2011) or evolutionary algorithms (Glass et al., 2006; Wang et al., 2010). Deep generative models learn to approximate underlying probability distributions from existing material data, and in turn, can be sampled from to generate novel materials based on the learned patterns.

Among deep generative models, diffusion models have been successful on a variety of data modalities relevant to the sciences, ranging from Partial Differential Equations (PDE) simulations (Lippe et al., 2023; Rozet & Louppe, 2023; Shysheya et al., 2024) to molecule generation (Hoogeboom et al., 2022; Xu et al., 2023; Cornet et al., 2024). Unlike molecules, crystalline materials consist of a periodic arrangement of atoms, typically described by a *unit cell*, a parallelepiped that serves as the fundamental building block repeated to tile the entire space. A unit cell is typically represented by three vectors that define its edges and the angles between them, along with the coordinates and species of the atoms inside it. It can be considered as a multi-modal data type, specifically a geometric graph combining discrete and continuous features (Joshi et al., 2023). Notably, the atomic positions are described by coordinates that lie on a hypertorus. Dealing with non-Euclidean data in the diffusion setting requires careful consideration: for example, restricting Brownian motion to a manifold requires incorporating its geometric structure to ensure trajectories remain on the manifold, typically using projection or specialized approximations (Lou et al., 2023). Current diffusion models for crystals either handle this by working in real space, through multi-graph representations (Xie et al., 2022), or on fractional coordinates (Jiao et al., 2023). In addition to this, the data distribution is governed by inherent symmetries, including permutation invariance (swapping atom indices

*Equal contribution  [1]Technical University of Denmark, Kongens Lyngby, Denmark  [2]Pioneer Center for Artificial Intelligence, Copenhagen, Denmark. Correspondence to: François Cornet `<frjc@dtu.dk>`, Federico Bergamin `<fedbe@dtu.dk>`.

*Proceedings of the 42$^{nd}$ International Conference on Machine Learning*, Vancouver, Canada. PMLR 267, 2025. Copyright 2025 by the author(s).

or lattice bases), translation invariance (shifting atom coordinates), and rotation invariance (rotating atom positions).

When operating on fractional coordinates, the main challenge is to ensure the periodic translation invariance of the learned distribution. This is usually enforced by parameterizing the score network with a periodic translation invariant architecture. However, while existing models have successfully demonstrated the potential of applying diffusion to crystalline materials generation, previous work has highlighted the existence of a mismatch between architecture and supervision signal, resulting in a suboptimal training objective (Lin et al., 2024). While the issue has been acknowledged in the literature, in this paper we show an alternative way to mitigate this mismatch that is effective at low levels of noise.

**Contributions** In this work, we introduce Kinetic Langevin Diffusion for Materials (KLDM), a novel diffusion model for crystalline materials generation, where the key innovation resides in the modeling of the fractional coordinates. Instead of resorting to Riemannian diffusion on the hypertorus directly as in previous work (Jiao et al., 2023), we generalize Trivialized Diffusion Model (TDM) (Zhu et al., 2024) to model symmetric distributions over coordinates. Using the structure of the torus, the diffusion process is offset to auxiliary Euclidean variables representing velocities. We propose a specific parameterization of the resulting diffusion process leading to faster training convergence and improved performance while mitigating a mismatch in the training objective of previous diffusion models for materials. Finally, we show that KLDM offers competitive performance on Crystal Structure Prediction (CSP) and De-novo Generation (DNG).

## 2. Background

### 2.1. Crystalline materials and related symmetries

In this section, we introduce the data modality we are interested in, along with the relevant symmetries.

**Unit cell** We are interested in learning the distribution of crystalline materials, described as the repetition of a unit cell in 3D-space. We describe a unit cell containing $K$ atoms as a fully connected geometric graph $\boldsymbol{x}$,

$$\boldsymbol{x} = (\boldsymbol{f}, \boldsymbol{l}, \boldsymbol{a}), \tag{1}$$

where $\boldsymbol{f} = (\boldsymbol{f}_1, \ldots, \boldsymbol{f}_K) \in [0, 1)^{3 \times K}$ denotes the fractional coordinates, $\boldsymbol{l} = (\boldsymbol{l}_1, \boldsymbol{l}_2, \boldsymbol{l}_3) \in \mathbb{R}^{3 \times 3}$ refers to the lattice vectors of the unit cell, and $\boldsymbol{a} = (a_1, \ldots, a_K) \in \mathbb{Z}^K$ encodes the chemical composition.

The infinite periodic structure can be represented as,

$$\{(\boldsymbol{a}', \boldsymbol{f}') | \boldsymbol{a}' = \boldsymbol{a}; \boldsymbol{f}' = \boldsymbol{f} + \boldsymbol{k}\mathbf{1}^\top, \boldsymbol{k} \in \mathbb{Z}^3\}, \tag{2}$$

where $\mathbf{1}$ is a vector of ones with size $K$, and $\boldsymbol{k} = (k_1, k_2, k_3)$ translates the unit cell to tile the entire space. The lattice vectors $\boldsymbol{l}$ can also be compactly represented by a 6-dimensional vector consisting of the three lattice vector lengths and their interior angles, such that the representation is invariant to rotation.

**Tasks** Let $p(\boldsymbol{x})$ denote the true data distribution over the unit cells, and let $q(\boldsymbol{x}) = \frac{1}{N} \sum_{i=1}^{N} \delta_{\boldsymbol{x}_i}$ be the empirical data distribution defined by the $N$ available samples. The goal of generative models is to learn an approximate model $p_\theta(\boldsymbol{x})$ of $p(\boldsymbol{x})$ using $q(\boldsymbol{x})$. In the realm of crystalline materials generation, there are two tasks of interest.

**Crystal Structure Prediction (CSP)** aims at finding low-energy atomic arrangements $(\boldsymbol{f}, \boldsymbol{l})$, *i.e.* stable structures, for a given atomic composition $\boldsymbol{a}$. This is framed as a conditional generation task where a model $p_\theta(\boldsymbol{f}, \boldsymbol{l} | \boldsymbol{a})$ is trained using samples $\{(\boldsymbol{f}, \boldsymbol{l}, \boldsymbol{a})\}_{i=1}^{N}$ to approximate the true conditional distribution $p(\boldsymbol{f}, \boldsymbol{l} | \boldsymbol{a})$. Given a specific atomic composition $\boldsymbol{a}$, the model is used to generate possible coordinates $\boldsymbol{f}$ and lattice parameters $\boldsymbol{l}$.

**De-novo Generation (DNG)** aims at discovering novel and stable materials. The goal is to generate samples from $p(\boldsymbol{f}, \boldsymbol{l}, \boldsymbol{a})$. We approximate the distribution by training a generative model $p_\theta(\boldsymbol{f}, \boldsymbol{l}, \boldsymbol{a})$ on samples $\{(\boldsymbol{f}, \boldsymbol{l}, \boldsymbol{a})_i\}_{i=1}^{N}$ and evaluate the samples generated by $p_\theta$.

**Symmetries of crystalline materials** Given a symmetry group $G$, a distribution is $G$-invariant if for any group element $g \in G$, $p(g \cdot \boldsymbol{x}) = p(\boldsymbol{x})$, with $\cdot$ denoting the group action. A conditional distribution is $G$-equivariant if for any $g \in G$, $p(g \cdot \boldsymbol{x} | g \cdot \boldsymbol{y}) = p(\boldsymbol{x} | \boldsymbol{y})$.

A number of transformations leave a material $\boldsymbol{x}$ unchanged, meaning the *true* data distribution has inherent symmetries:

Permutation of atom indices
$$p(\boldsymbol{f}, \boldsymbol{l}, \boldsymbol{a}) = p(g \cdot \boldsymbol{f}, \boldsymbol{l}, g \cdot \boldsymbol{a}), \qquad \forall g \in S_K; \tag{3}$$

Rotation of lattice vectors
$$p(\boldsymbol{f}, \boldsymbol{l}, \boldsymbol{a}) = p(\boldsymbol{f}, g \cdot \boldsymbol{l}, \boldsymbol{a}), \qquad \forall g \in \mathrm{SO}(3); \tag{4}$$

Permutation of the lattice basis
$$p(\boldsymbol{f}, \boldsymbol{l}, \boldsymbol{a}) = p(g \cdot \boldsymbol{f}, g \cdot \boldsymbol{l}, \boldsymbol{a}), \qquad \forall g \in S_3; \tag{5}$$

Periodic translation of fractional coordinates
$$p(\boldsymbol{f}, \boldsymbol{l}, \boldsymbol{a}) = p(g \cdot \boldsymbol{f}, \boldsymbol{l}, \boldsymbol{a}) \qquad \forall g \in \mathbb{T}^3 \cong \mathbb{R}^3 / \mathbb{Z}^3. \tag{6}$$

We want our model, $p_\theta$, to inherit these symmetries. Eq. (3) is naturally addressed by using graph neural networks, combined with a factorized prior distribution with no dependency on the index. The combination of fractional coordinates and rotation-invariant lattice representations addresses Eq. (4). Specific neural network architectures or additional

loss terms can address Eq. (5)—see *e.g.* (Lin et al., 2024). Remaining is to handle Eq. (6), we detail how it can be done in Section 2.3.

## 2.2. Diffusion models

Diffusion models (Sohl-Dickstein et al., 2015; Ho et al., 2020; Song et al., 2021) are generative models that learn distributions through a hierarchy of latent variables, corresponding to corrupted versions of the data at increasing noise scales. Diffusion models consist of a forward and a reverse (generative) process. The forward process perturbs samples from the data distribution over time through noise injection, resulting in a trajectory of increasingly noisy latent variables $(\boldsymbol{x}_t)_{t\in[0,T]}$. Given an initial condition, $\boldsymbol{x}_0 \sim p_0(\boldsymbol{x}) = p_{\text{data}}(\boldsymbol{x})$, the conditional distribution of $(\boldsymbol{x}_t)_{t\in[0,T]}$ can be characterized by a Stochastic Differential Equation (SDE),

$$\mathrm{d}\boldsymbol{x}_t = f(t)\boldsymbol{x}_t\,\mathrm{d}t + g(t)\,\mathrm{d}\boldsymbol{w}_t, \tag{7}$$

where $\boldsymbol{x}_t \in \mathbb{R}^d$ denotes the latent variable at time $t$, $f(t)$ and $g(t)$ are scalar function of time $t$, and $\boldsymbol{w}_t$ is a standard Wiener process in $\mathbb{R}^d$. Due to the linearity of the drift term, for any $t \geq 0$, the corresponding transition kernel admits a closed-form expression (Särkkä & Solin, 2019). For instance, in the Variance-Preserving SDE (VP-SDE) setting (Song et al., 2021), where $f(t) = -\frac{1}{2}\beta(t)$ and $g(t) = \sqrt{\beta(t)}$ for a fixed schedule $\beta(t)$, the kernel writes $p_{t|0}(\boldsymbol{x}_t|\boldsymbol{x}_0) = \mathcal{N}(\boldsymbol{x}_t|\alpha_t\boldsymbol{x}_0, \sigma_t^2\mathbb{I})$ with $\alpha_t = \exp(-0.5\int_0^t \beta(s)\mathrm{d}s)$ and $\sigma_t^2 = 1 - \exp(\int_0^t \beta(s)\mathrm{d}s)$, and the process defined by Eq. (7) converges geometrically from a low-variance Gaussian distribution centered around the data to the standard Gaussian distribution $p_T(\boldsymbol{x}_t) = \mathcal{N}(\boldsymbol{x}_t|\boldsymbol{0}, \mathbb{I})$, which can be therefore interpreted as an uninformative prior distribution.

The time-reversal of Eq. (7) is another diffusion process described by the following reverse-time SDE (Anderson, 1982),

$$\mathrm{d}\boldsymbol{x}_t = \left[f(t)\boldsymbol{x}_t - g^2(t)\nabla_{\boldsymbol{x}_t}\log p_t(\boldsymbol{x})\right]\mathrm{d}t + g(t)\,\mathrm{d}\hat{\boldsymbol{w}}_t, \tag{8}$$

with $p_t(\boldsymbol{x}_t)$ being the density of $\boldsymbol{x}_t$ and $\mathrm{d}\hat{\boldsymbol{w}}_t$ is a time-reversed Wiener process. Sampling from the prior distribution $p_T(\boldsymbol{x}_t)$, and simulating Eq. (8) results in a sample from $p(\boldsymbol{x})$. In practice, the score $\nabla_{\boldsymbol{x}_t}\log p_t(\boldsymbol{x})$ is not available and it is approximated using a *score network* $s_\theta(\boldsymbol{x}_t, t)$, whose parameters $\theta$ are trained via Denoising Score Matching (DSM),

$$\mathcal{L}_{\text{DSM}} = \mathbb{E}_{\lambda(t)}\left[\left\|s_\theta(\boldsymbol{x}_t, t) - \nabla_{\boldsymbol{x}_t}\log p_{t|0}(\boldsymbol{x}_t|\boldsymbol{x}_0)\right\|_2^2\right], \tag{9}$$

where $\lambda(t)$ is a positive time-dependent weighting function and the expectation is taken over the joint distribution $t \sim \mathcal{U}[0,1]$, $\boldsymbol{x}_0 \sim p(\boldsymbol{x})$, and $\boldsymbol{x}_t \sim p_{t|0}(\boldsymbol{x}_t|\boldsymbol{x}_0)$.

## 2.3. Existing diffusion models for fractional coordinates

As previously mentioned, the fractional coordinates define a hypertorus $\boldsymbol{f} \in [0, 1)^{3\times K} \cong \mathbb{T}^{3\times K}$. Most existing work has built upon DIFFCSP (Jiao et al., 2023), which leverages the score-based framework of Song et al. (2021) extended to Riemannian manifolds (De Bortoli et al., 2022; Jing et al., 2022).

In what follows, we present the key ingredients of DIFFCSP for modeling fractional coordinates.

**Transition kernel** DIFFCSP implements a specific case of Eq. (7)—*i.e.* Variance-Exploding SDE (VE-SDE), with $f(t) = 0$ and $g(t) = \sqrt{\mathrm{d}\sigma^2(t)/\mathrm{d}t}$, where $\sigma(t) = \sigma_{\min}^{1-t}\sigma_{\max}^t$ with $\sigma_{\min}$ and $\sigma_{\max}$ being hyperparameters.

In practice, as sampling noisy fractional coordinates $\boldsymbol{f}_t$ given $\boldsymbol{f}_0$ using a normal distribution does not capture the bounded and cyclical nature of $p(\boldsymbol{f})$, the solution consists in first sampling from the normal distribution $\hat{\boldsymbol{f}}_t \sim \mathcal{N}\left(\boldsymbol{f}_0, \sigma^2(t)\right)$ and then wrapping the samples $\boldsymbol{f}_t = w(\hat{\boldsymbol{f}}_t)$—where $w(\cdot) = \cdot - \lfloor\cdot\rfloor$ is the wrapping function.

The transition kernel corresponding to this two-step procedure corresponds to a wrapped normal distribution,

$$p_{t|0}(\boldsymbol{f}_t|\boldsymbol{f}_0) \propto \sum_{\boldsymbol{k}\in\mathbb{Z}^{3\times K}} \exp\left(-\frac{\|\boldsymbol{f}_t - \boldsymbol{f}_0 + \boldsymbol{k}\|^2}{2\sigma^2(t)}\right), \tag{10}$$

whose gradient can then be approximated by truncating the series, allowing for training using denoising score-matching. Due to the wrapping operation, Eq. (10) converges to a uniform distribution over the hypertorus as $t \to T$.

**Denoising score-matching** Given the transition kernel $p_{t|0}(\boldsymbol{f}_t|\boldsymbol{f}_0)$, the approximate score function can be optimized by minimizing a denoising score-matching objective, which at time $t$ writes

$$\mathcal{L}_{\boldsymbol{f}_t} = \mathbb{E}_{\boldsymbol{f}_0, \boldsymbol{f}_t}\left[\lambda(t)\left\|\nabla_{\boldsymbol{f}_t}\log p_{t|0}(\boldsymbol{f}_t|\boldsymbol{f}_0) - s_\theta^{\boldsymbol{f}}(\boldsymbol{x}_t, t)\right\|_2^2\right], \tag{11}$$

where $\boldsymbol{f}_0 \sim p_0(\boldsymbol{f}), \boldsymbol{f}_t \sim p_{t|0}(\boldsymbol{f}_t|\boldsymbol{f}_0)$, $s_\theta^{\boldsymbol{f}}(\boldsymbol{x}_t, t)$ denotes the output of the score network corresponding to fractional coordinates, and $\lambda(t) = 1/\mathbb{E}_{\boldsymbol{f}_t \sim p_{t|0}(\boldsymbol{f}_t|\boldsymbol{f}_0)}\left[\|\nabla_{\boldsymbol{f}_t}\log p_{t|0}(\boldsymbol{f}_t|\boldsymbol{f}_0)\|_2^2\right]$ is a (precomputable) scale factor ensuring that the loss magnitude is constant in expectation across time (Jing et al., 2022; Jiao et al., 2023).

**Invariant approximate distribution** As introduced in Eq. (6), the true data distribution $p(\boldsymbol{x})$, with $\boldsymbol{x} = (\boldsymbol{f}, \boldsymbol{l}, \boldsymbol{a})$ as defined in Section 2.1, is periodic translation invariant—*i.e.* unit cells equivalent up to periodic translation shall be equally likely. In DIFFCSP (Jiao et al., 2023), the learned distribution $p_\theta(\boldsymbol{x})$ is made invariant by combining an invariant prior - *i.e.* uniform distribution on the hypertorus - with an approximate reverse process that is equivariant to periodic

translation—*i.e.* $p_\theta(\boldsymbol{x}_{t-1}|\boldsymbol{x}_t) = p_\theta(g \cdot \boldsymbol{x}_{t-1}|g \cdot \boldsymbol{x}_t), \forall g \in \mathbb{R}^3/\mathbb{Z}^3$. In practice, this is realized by parameterizing the score using a neural network that is invariant to periodic translations—*i.e.* $s_\theta^{\boldsymbol{f}}(\boldsymbol{x}_t, t) = s_\theta^{\boldsymbol{f}}(g \cdot \boldsymbol{x}_t, t), \forall g \in \mathbb{R}^3/\mathbb{Z}^3$. If the learned score $s_\theta^{\boldsymbol{f}}(\boldsymbol{x}_t, t)$ correctly approximates the score of the true target distribution, then we are guaranteed to model a distribution $p_\theta(\boldsymbol{x})$ that exhibits the same symmetries as $p(\boldsymbol{x})$ from which new samples can be drawn.

# 3. Kinetic Langevin Diffusion for Materials generation

We now introduce Kinetic Langevin Diffusion for Materials (KLDM), and particularize our exposition to the fractional coordinates $\boldsymbol{f} \in [0,1)^{3 \times K} \cong \mathbb{T}^{3 \times K}$, used to define the positions of atoms inside the unit cell.

Given the isomorphism between the torus $\mathbb{T}$ and SO(2)[1], the fractional coordinates can alternatively be described as a collection of $2 \times 2$ rotation matrices. We denote this alternative representation $\hat{\boldsymbol{f}}$. Since $\hat{\boldsymbol{f}}$ is defined on a direct product of connected compact Lie groups[2], we propose to generalize Trivialized Diffusion Model (TDM) (Zhu et al., 2024) to operate on geometric graphs similar to those defined in Eq. (1). While TDM in principle allows for proper treatment of the fractional coordinates *out-of-the-box*, we find its direct application to crystalline materials generation to result in slow convergence and subpar performance. We consequently propose a set of modifications designed to facilitate faster convergence and enhance empirical results.

## 3.1. Trivialized Diffusion Model (TDM) for fractional coordinates

Building upon previous work on momentum-based optimization (Tao & Ohsawa, 2020) and sampling (Kong & Tao, 2023), TDM is specifically tailored to data defined on Lie groups, and exploits the particular group structure, specifically the left-trivialization operation, to effectively perform diffusion on the manifold via the Lie algebra. The main idea is to couple the variables of interest defined on the manifold with auxiliary variables representing velocities defined on the Lie algebra. More precisely, the fractional coordinates $\hat{\boldsymbol{f}}$, elements of the group $G = \text{SO}(2)^{3 \times K}$, are coupled with velocities $\hat{\boldsymbol{v}} \in \mathfrak{g}$ defined on the Lie algebra $\mathfrak{g}$. The latter corresponds to the tangent space $T_eG$ at the identity element of the group $e \in G$, and crucially can be thought of as a Euclidean space, $\mathfrak{g} \cong \mathbb{R}^{3 \times K}$. In the present setting, velocities

---

[1]We provide an intuitive explanation of this correspondence in Appendix B.1.

[2]A Lie group is a smooth manifold equipped with a group structure denoted by $G$ and smooth group operations. We provide a short informal introduction to Lie groups and Lie algebra in Appendix A.

---

$\hat{\boldsymbol{v}}$ are $2 \times 2$ skew-symmetric matrices, whose anti-diagonal element, $v$, is real-valued.

**Forward process** As the Lie algebra is isomorphic to Euclidean space, a standard diffusion process can be defined for $\hat{\boldsymbol{v}}$. Given its coupling with $\hat{\boldsymbol{f}}$ determined by left-trivialization (Zhu et al., 2024), the resulting forward process is defined as,

$$\begin{cases} \mathrm{d}\hat{\boldsymbol{f}}_t &= \hat{\boldsymbol{f}}_t \hat{\boldsymbol{v}}_t \, \mathrm{d}t, \\ \mathrm{d}\hat{\boldsymbol{v}}_t &= -\gamma \hat{\boldsymbol{v}}_t \, \mathrm{d}t + \sqrt{2\gamma} \, \mathrm{d}\boldsymbol{w}_t^{\mathfrak{g}}, \end{cases} \tag{12}$$

where the Ordinary Differential Equation (ODE) describes the time evolution of the fractional coordinates $\hat{\boldsymbol{f}}_t$ (that lie on a manifold) through a coupling with the velocity variables $\hat{\boldsymbol{v}}_t$. The latter evolve according to an SDE similar to that of Eq. (7), with constant drift $f(t) = -\gamma$ and constant volatility $g(t) = \sqrt{2\gamma}$, and where $\boldsymbol{w}_t^{\mathfrak{g}}$ denotes a standard Wiener process on the Lie algebra $\mathfrak{g}$. We note that, in principle, $f$ and $g$ could be generalized to time-dependent functions but that we consider constant here. In summary, Eq. (12) corrupts $\hat{\boldsymbol{f}}_t$ living on a hypertorus via a standard Euclidean diffusion process on the auxiliary variables, while guaranteeing that the trajectory $(\hat{\boldsymbol{f}}_t)_{t \in [0,T]}$ remains on the manifold at all times. Eq. (12) converges to a product of easy-to-sample distributions: a uniform distribution on $G$, and a standard Normal distribution on $\mathfrak{g}$.

**Reverse (generative) process** The time-reversal of Eq. (12) is given by

$$\begin{cases} \mathrm{d}\hat{\boldsymbol{f}}_t = \hat{\boldsymbol{f}}_t \hat{\boldsymbol{v}}_t \mathrm{d}t, \\ \mathrm{d}\hat{\boldsymbol{v}}_t = \left[ -\gamma\hat{\boldsymbol{v}}_t - 2\gamma\nabla_{\hat{\boldsymbol{v}}_t} \log p_t(\hat{\boldsymbol{f}}_t, \hat{\boldsymbol{v}}_t) \right] \mathrm{d}t \\ \qquad\qquad\qquad\qquad + \sqrt{2\gamma} \, \mathrm{d}\hat{\boldsymbol{w}}_t^{\mathfrak{g}}, \end{cases} \tag{13}$$

where $t$ flows backwards and $\nabla_{\hat{\boldsymbol{v}}_t} \log p_t(\hat{\boldsymbol{f}}_t, \hat{\boldsymbol{v}}_t)$ denotes the (true) score. The latter is unavailable, but can be approximated with a neural network $s_\theta(\hat{\boldsymbol{f}}_t, \hat{\boldsymbol{v}}_t, t)$ trained using DSM as in Eq. (11). However, we stress that the gradient is now with respect to $\hat{\boldsymbol{v}}_t$—*i.e.* an Euclidean variable, in contrast to Eq. (11) where it was with respect to $\boldsymbol{f}_t$—see Eq. (17). As in Eq. (12) there is no direct noise in the forward dynamics of $\hat{\boldsymbol{f}}_t$, its time-reversal in Eq. (13) does not need score-based correction and corresponds to the reverse ODE.

Practically, we split the vector field of the generative dynamics in Eq. (13) into two vector fields: one for $\hat{\boldsymbol{f}}_t$ where $\hat{\boldsymbol{v}}_t$ is considered fixed—this is a simple linear ODE; and one for $\hat{\boldsymbol{v}}_t$ where $\hat{\boldsymbol{f}}_t$ is considered fixed—this is a semi-linear SDE, as usually encountered in diffusion models.

**Discretized update of the fractional coordinates** The ODEs in Eqs. (12) and (13) describe the dynamics associated with the fractional coordinates. Since they lie on a manifold, integrating the dynamics from time $t$ to $t + \mathrm{d}t$ involves solving a Riemannian exponential map. Intuitively, this

operation generalizes the concept of moving on a straight line to the manifold case. By considering an initial velocity $\hat{v}_t$, an initial position $\hat{f}_t$ and the step-size $dt$, the update of the positions can be written as

$$\hat{f}_{t+dt} = \exp_{\hat{f}_t}(\hat{f}_t\hat{v}_t dt) = \hat{f}_t\exp m(\hat{v}_t dt), \quad (14)$$

where the matrix multiplication $\hat{f}_t\hat{v}_t dt$ returns the tangent vector in $\hat{f}_t$, and the matrix exponential is the exact solution of the exponential map (Zhu et al., 2024).

The formalism previously introduced considered $\hat{f}$ as a collection of rotation matrices. However, the usual neural networks used to parameterize the score have been designed to process fractional coordinates (Jiao et al., 2023; Lin et al., 2024). We therefore want to work on the hypertorus $\mathbb{T}^{3\times K}$ directly, using the original representation of the fractional coordinates $f$. Concretely, if we consider the exponential map defined in Eq. (14) and that $\hat{f}^i$ is a single rotation matrix by an angle $\theta \in [-\pi, \pi]$, the matrix exponential corresponds to a periodic translation as follows,

$$\hat{f}^i\exp m(\hat{v}^i dt) \to w(\theta^i + v^i dt), \quad (15)$$

where $\hat{v}^i$ is a skew-symmetric matrix and $v^i$ is its anti-diagonal element, while $w(\cdot) = \text{atan2}\big(\sin(\cdot), \cos(\cdot)\big)$ is the wrapping function with $\text{atan2}$ denoting the signed $\text{atan}$ function—see Appendix B.2 for more details. Since $\theta^i$ in Eq. (15) corresponds to a scaled version of the corresponding coordinate $f^i$, we continue the presentation with all operations expressed in terms of $f$.

**Transition kernel (Zhu et al., 2024)** The transition kernel corresponding to the forward dynamics of Eq. (12) writes

$$p_{t|0}(f_t, v_t|f_0, v_0) = \quad (16)$$
$$\text{WN}(r_t|\mu_{r_t}, \sigma_{r_t}^2\mathbf{I}) \cdot \mathcal{N}_v(v_t|\mu_{v_t}, \sigma_{v_t}^2\mathbf{I}),$$

where $r_t = \text{logm}(f_0^{-1}f_t)$, $\text{WN}(\cdot|\mu_{r_t}, \sigma_{r_t}^2\mathbf{I})$ denotes the density of the Wrapped Normal distribution with mean $\mu_{r_t} = \frac{1-e^{-t}}{1+e^{-t}}(v_t + v_0)$ and variance $\sigma_{r_t}^2 = 2t + \frac{8}{e^t+1} - 4$, while $\mathcal{N}_v$ is a Gaussian distribution with $\mu_{v_t} = e^{-t}v_0$ and $\sigma_{v_t}^2 = 1 - e^{-2t}$.

Practically, noisy samples $f_t$ can be obtained from the transition kernel in Eq. (16) as follows. First, we sample the auxiliary variables $v_t$. We then sample $r_t = \text{logm}(f_0^{-1}f_t)$. Finally, $f_t$ is obtained as $f_t = f_0\exp m(r_t)$.

The joint distribution in Eq. (16) evolves from the product $p(f) \cdot p(v_0)$, to a tractable limiting distribution that is the product of a uniform distribution over $\mathbb{T}^{3\times K}$ in $f$ and a standard Normal in $v$.

**Denoising score-matching target** As the diffusion process only acts on the (Euclidean) velocity variables, we only need to compute the score of Eq. (16) with respect to the velocity

variables, $v_t$, unlike the objective presented in Eq. (11) where the gradient was instead computed with respect to the coordinates $f_t$. It writes,

$$\nabla_{v_t}\log p_{t|0} = \nabla_{\mu_{r_t}}\log\text{WN}(r_t|\mu_{r_t}, \sigma_{r_t}^2\mathbf{I})\frac{\partial\mu_{r_t}}{\partial v_t} \quad (17)$$
$$+ \nabla_{v_t}\log\mathcal{N}_v(v_t|\mu_{v_t}, \sigma_{v_t}^2\mathbf{I}),$$

where we shortened $p_{t|0}(f_t, v_t|f_0, v_0)$ as $p_{t|0}$. Eq. (17) is the target used to train our score network with the DSM objective defined in Eq. (9).

### 3.2. Modeling choices

**Initial zero velocities** Defining $p(v_0)$ is a design choice, and we find that setting the initial velocities to zero, *i.e.* $p(v_0) = \delta(v_0)$, to be beneficial. This intuitively corresponds to considering coordinates $f$ initially at rest, at time $t = 0$, with noise gradually propagating from the velocities to the coordinates over the course of the diffusion process. A similar benefit was also observed by Dockhorn et al. (2022), for small initial velocities.

**Simplified score parameterization** While predicting the full score in Eq. (17) directly is possible, alternative parameterizations exist—similar to what is done in standard diffusion models. By carefully inspecting the score expression, we note that the target writes as a sum of two terms, where some of the quantities are known,

$$\nabla_{v_t}\log p_{t|0}(f_t, v_t|f_0, v_0) \quad (18)$$
$$= \frac{1-e^{-t}}{1+e^{-t}}\nabla_{\mu_{r_t}}\text{WN}(r_t|\mu_{r_t}, \sigma_{r_t}^2\mathbf{I}) - \frac{\varepsilon_v}{\sigma_{v_t}},$$

with $\varepsilon_v$ denoting the reparameterization noise sampled to obtain $v_t$. When considering zero initial velocities, *i.e.* $v_0 = 0$, we have that $\mu_{v_t} = 0, \forall t$. Using the relationship, $v_t = \mu_{v_t} + \sigma_{v_t}\varepsilon_v$, the second term in Eq. (18) can be computed in closed-form—as we have $\varepsilon_v = v_t/\sigma_{v_t}$, and does not need to be learned. We are then left with a single unknown term, and the simplified parameterization writes,

$$s_\theta^v(x_t, t) = \frac{1-e^{-t}}{1+e^{-t}}s_\theta^f(x_t, t) - \frac{v_t}{\sigma_{v_t}^2}, \quad (19)$$

where superscript $f$ in $s_\theta^f(x_t, t)$ refers to the output of the score network corresponding to fractional coordinates. More details are provided in Appendix C.

Empirically, we find the combination of the zero initial velocities and the simplified parameterization to be beneficial for convergence and performance, as discussed in Appendix E.

**Architecture of the score network** As imposed by the symmetries inherent to the target distribution, we parameterize our score network, $s_\theta(x_t, t)$, using a graph neural network

architecture, with a backbone similar to that of previous work (Jiao et al., 2023). The sole difference is that the score network now takes the auxiliary variables, $v_t$, as additional inputs. The network is made invariant to periodic translation of $f_t$, by featurizing pairwise differences of fractional coordinates with periodic functions of different frequencies, or similarly with shortest distances on the flat torus. More details about the architecture are provided in Appendix I.

**Velocity fields with zero net translation** Due to the translational symmetry of the target distribution - and consequently the translation invariance of the score parametrization - we want to prevent velocities $(v_t)_{t \in [0, T]}$ from inducing a net overall translation of the system at every time step. Since velocities are defined on the Lie algebra and can therefore be viewed as Euclidean variables, we consider velocity fields living in the mean-free linear subspace. This is, in essence, similar to subtracting the center of gravity when working with molecules in Euclidean space (Xu et al., 2022; Hoogeboom et al., 2022), except that it now takes place on the velocities defined on the Lie algebra.

In practice, this implies that the distribution of velocities, $\mathcal{N}_v$ in Eq. (16), corresponds to a projected Normal where all samples are mean-free—that is, they satisfy $\sum_k^K v_k = \mathbf{0}$, to account for the constraint on the velocities. We also remove the mean from the sampled $r_t$, and project the score in Eq. (17), to ensure that it does not introduce a net translation. Finally, the corresponding velocity score output $s_\theta^v(x_t, t)$ in Eq. (19) - and hence $s_\theta^f(x_t, t)$ - is constructed to be mean-free as well.

Ideally, we would like to preserve the group element $g$ used to represent the clean sample $f_0$, since the score network is insensitive to translation. While restricting the velocity fields to the mean-free subspace prevents global drift, it does not always ensure that the noisy coordinates, $f_t$, maintain the *mean*[3] of the original sample, $f_0$.

However, we empirically observe that the mean tends to be preserved at moderate noise levels (see Figs. 2 and 3). This helps mitigate the potential mismatch that exists between the translation-invariant score parameterization and the non-invariant training target.

This issue has been pointed out in previous. Lin et al. (2024), for example, proposed a so-called *Periodic CoM-free Noising* scheme, in which the noise is carefully designed to ensure a translation-invariant training target. The resulting transition kernel is formulated as a von Mises distribution and estimated numerically via Monte Carlo simulations.

We refer the reader to Appendix D for a longer discussion

and a more detailed analysis.

### 3.3. Other modalities

**Lattice vectors** We represent lattices $l$ as 6-dimensional vectors, collecting side lengths and angles. We follow previous work (Jiao et al., 2023; Lin et al., 2024) and rely on standard Euclidean diffusion (VP-SDE) as defined in Eqs. (7) and (8), with drift and diffusion functions defined by a linear schedule.

**Compositions** In the DNG setting, we employ three distinct representations for atomic compositions $a$, as per previous work: continuous diffusion using one-hot encoding (Jiao et al., 2023); continuous diffusion on analog-bits (Chen et al., 2023; Miller et al., 2024); and discrete (absorbing) diffusion (Austin et al., 2021; Jiao et al., 2023).

## 4. Related work

Early deep generative models for crystal generation leveraged image representations (Hoffmann et al., 2019; Court et al., 2020), ad-hoc frequency space representations (Ren et al., 2022) or 3D coordinates without considering their geometric nature (Nouira et al., 2018; Kim et al., 2020; Yang et al., 2024b).

Since then, several works have leveraged diffusion models operating on geometric graphs. The seminal approach of Xie et al. (2022) initially worked in *real space*, and resorted to multi-graphs to account for periodicity (Xie & Grossman, 2018). The diffusion process was limited to coordinates, with fixed lattice parameters and composition predicted by a VAE. While such a model has shown practical usefulness, *e.g.* to find novel 2D materials (Lyngby & Thygesen, 2022), more recent diffusion models instead defined a (more flexible) joint diffusion process for coordinates, lattice structure, and atom types (Jiao et al., 2023; Zeni et al., 2025), and accounted for periodicity by operating on fractional coordinates. Miller et al. (2024) generalized Riemannian flow matching (Lipman et al., 2023; Chen & Lipman, 2024) to the same setup. Sriram et al. (2024) further leveraged the flexibility of flow matching and used a fine-tuned LLM as a base distribution. Others (Flam-Shepherd & Aspuru-Guzik, 2023; Gruver et al., 2024; Antunes et al., 2024) also trained or fine-tuned LLMS on text representation of materials and demonstrated the ability of such models to generate valid descriptions, sometimes outperforming domain-specific methods.

Finally, a recent line of work (Jiao et al., 2024; Levy et al., 2024) has sought to exploit space-group information to restrict generation to the smallest asymmetric part of the unit cell only, including for disordered materials (Petersen et al., 2025).

---

[3]In this case, we are referring to the mean defined on a manifold and not the Euclidean mean. See Appendix D for a more precise definition.

Table 1: **Crystal Structure Prediction (CSP) results**. Baseline results are extracted from the respective papers. @ indicates the number of samples considered to evaluate the metrics, *e.g.* @20 indicates the best of 20. Error bars for @1 represent the standard deviation over the mean at sampling time across 20 different seeds. Most notably, KLDM compares favorably to the competing models, achieving performance comparable to or better than state-of-the-art (SOTA) methods.

| MODEL | PEROV-5 | | MP-20 | | MPTS-52 | |
|---|---|---|---|---|---|---|
| | MR [%] ↑ | RMSE ↓ | MR [%] ↑ | RMSE ↓ | MR [%] ↑ | RMSE ↓ |
| | | | METRICS @ 1 | | | |
| CDVAE | 45.31 | 0.1138 | 33.90 | 0.1045 | 5.34 | 0.2106 |
| DIFFCSP (PC) | 52.02 | 0.0760 | 51.49 | 0.0631 | 12.19 | 0.1786 |
| EQUICSP (PC) | 52.02 | 0.0707 | 57.59 | $\underline{0.0510}$ | 14.85 | **0.1169** |
| FLOWMM | **53.15** | 0.0992 | 61.39 | 0.0566 | 17.54 | 0.1726 |
| **KLDM-$\varepsilon$** (EM) | $\underline{53.14}_{\pm.6}$ | $0.0758_{\pm.002}$ | $61.72_{\pm.2}$ | $0.0686_{\pm.001}$ | $17.71_{\pm.3}$ | $0.2023_{\pm.005}$ |
| **KLDM-$\varepsilon$** (PC) | $52.72_{\pm.8}$ | $\underline{0.0678}_{\pm.002}$ | $\underline{65.37}_{\pm.1}$ | $\mathbf{0.0455}_{\pm.001}$ | $\underline{21.46}_{\pm.2}$ | $0.1339_{\pm.002}$ |
| **KLDM-$x_0$** (EM) | $52.44_{\pm.7}$ | $0.0698_{\pm.002}$ | $62.92_{\pm.2}$ | $0.0833_{\pm.002}$ | $21.13_{\pm.2}$ | $0.1800_{\pm.003}$ |
| **KLDM-$x_0$** (PC) | $52.14_{\pm.9}$ | $\mathbf{0.0647}_{\pm.002}$ | $\mathbf{65.83}_{\pm.2}$ | $0.0517_{\pm.001}$ | $\mathbf{23.93}_{\pm.2}$ | $\underline{0.1276}_{\pm.002}$ |
| | | | METRICS @ 20 | | | |
| CDVAE | 88.51 | 0.0464 | 66.95 | 0.1026 | 20.79 | 0.2085 |
| DIFFCSP (PC) | 98.60 | **0.0128** | 77.93 | 0.0492 | 34.02 | 0.1749 |
| FLOWMM | 98.60 | 0.0328 | 75.81 | 0.0479 | 34.05 | 0.1813 |
| **KLDM-$\varepsilon$** (EM) | **99.97** | $\underline{0.0152}$ | **83.68** | 0.0532 | $\underline{39.04}$ | 0.1865 |
| **KLDM-$\varepsilon$** (PC) | $\underline{99.94}$ | 0.0226 | 81.08 | **0.0440** | **39.81** | $\underline{0.1462}$ |
| **KLDM-$x_0$** (EM) | 99.89 | 0.0186 | $\underline{82.94}$ | 0.0575 | 37.77 | 0.1673 |
| **KLDM-$x_0$** (PC) | 99.92 | 0.0255 | 80.18 | $\underline{0.0453}$ | 37.10 | **0.1394** |

## 5. Experimental results

### 5.1. Settings

**Tasks and Datasets** We now evaluate KLDM on the two tasks outlined in Section 2.1: Crystal Structure Prediction (CSP) and De-novo Generation (DNG).

We follow previous work (Jiao et al., 2023) and evaluate KLDM across 4 datasets: PEROV-5 (Castelli et al., 2012) including perovskite materials with 5 atoms per unit cell (ABX$_3$), all sharing the same structure but differing in composition; MP-20 including almost all experimentally stable materials from the Materials Project (Jain et al., 2013), with unit cells containing at most 20 atoms; and MPTS-52 also extracted from the Materials Project (Jain et al., 2013), with unit cells containing up to 52 atoms.

We describe the exact experimental setting in Appendix I, and release a public code repository with our implementation of KLDM.

**Sampling schemes** As DIFFCSP (Jiao et al., 2023) and EQUICSP (Lin et al., 2024) both rely on a Predictor-Corrector (PC) integrator (Song et al., 2021), we consider two different integration schemes for KLDM: the first combines Euler–Maruyama (EM) for the lattice parameters with an exponential integrator for the velocities, whereas the second applies a PC scheme to the velocities and EM for the other modalities. We provide detailed pseudo-code for the training and sampling procedures in Appendix H.

### 5.2. Crystal Structure Prediction (CSP) task

**Model setup** We present two different models that differ in terms of the score parameterization for the lattice parameters $l$. We consider a model that uses the $\varepsilon$-parameterization (KLDM-$\varepsilon$) and one using the $x_0$ parameterization (KLDM-$x_0$). For the latter, we also standardize the value of the lattice parameters following (Miller et al., 2024).

**Metrics** We follow the evaluation procedure of Xie et al. (2022) to assess the quality of the structures generated by KLDM. We report Match Rate (MR), measuring the proportion of reconstructions from $q_\theta(f, l|a)$ that are satisfactorily close to the ground truth structures as per StructureMatcher (Ong et al., 2013); and Root-Mean-Square-Error (RMSE), quantifying the RMSE between coordinates of matching reconstructions and ground truth structures.

**Baselines** We compare KLDM to the following recent generative models: CDVAE (Xie et al., 2022), DIFFCSP (Jiao et al., 2023), EQUICSP (Lin et al., 2024), and FLOWMM (Miller et al., 2024).

**Results** We perform CSP on all datasets and present the corresponding results in Table 1. We note that this task constitutes an ideal test bench for measuring the effect of the novel treatment of the fractional coordinates specific to KLDM. On the simpler PEROV-5 dataset, KLDM performs on par with the compared models for the @1 experiment, and

Table 2: **De-novo Generation (DNG) results on MP-20**. Stability metrics are evaluated using MATTERGEN's pipeline (Zeni et al., 2025) with MatterSim-v1-1M (Yang et al., 2024a). Baseline results are taken from MATTERGEN's own benchmark. KLDM was trained on the original MP-20 dataset, whereas MATTERGEN-MP* and DIFFCSP* were trained on a re-optimized version. We report average and standard deviation across 3 sampling seeds. Notably, KLDM with analog-bit or discrete diffusion outperforms DIFFCSP in terms of RMSD, energy above the hull, and stability, while being slightly subpar on RMSD.

| | RMSD [Å] ↓ | AVG. ABOVE HULL [eV/atom] ↓ | STABLE [%] ↑ | S.U.N. [%] ↑ |
|---|---|---|---|---|
| MATTERGEN-MP* | 0.147 | 0.201 | 47.05 | 25.76 |
| DIFFCSP* | 0.413 | 0.189 | 41.25 | 20.13 |
| **KLDM-$x_0$ (C)** | 0.371 $_{\pm.01}$ | 0.269 $_{\pm.01}$ | 38.62 $_{\pm.1}$ | 16.67 $_{\pm.1}$ |
| **KLDM-$x_0$ (C-AB)** | 0.296 $_{\pm.01}$ | 0.187 $_{\pm.01}$ | 49.84 $_{\pm.1}$ | 17.91 $_{\pm.1}$ |
| **KLDM-$x_0$ (D)** | 0.283 $_{\pm.01}$ | 0.155 $_{\pm.01}$ | 59.21 $_{\pm.1}$ | 18.52 $_{\pm.1}$ |

yields improved results for @20. Notably, the PC sampler does not enhance sample quality on this dataset. In contrast, on the more realistic datasets MP-20 and MPTS-52, KLDM already yields competitive MR for the EM sampler. The PC sampler further improves performance, improving MR and reducing RMSE. Additional gains are achieved on these datasets by standardizing the lattice parameters and adopting the $x_0$-parameterization for the lattice parameters.

For completeness, we report results on the CARBON-24 dataset in Appendix F, Table 4. The @1 task is inherently ill-defined, as the dataset consists exclusively of carbon atoms, allowing multiple distinct structures to satisfy the same composition. On the @20 task, KLDM compares favorably to baselines.

**5.3. De-novo Generation (DNG) task**

**Model setup** As described in Section 3.3, the CSP model formulation can be readily extended to the DNG task, by introducing an additional diffusion process operating on the atom types, $a$. To sample from $p_\theta(x)$, we first sample the number of atoms $K$ in the unit cell from the empirical distribution observed in the training set, *i.e.* from $p_\theta(x|K)p(K)$ (Hoogeboom et al., 2022).

For the lattice parameters $l$, we adopt the KLDM-$x_0$ variant, as it demonstrated the best performance in the CSP task on the larger datasets. Similarly, we use the PC sampler as it consistently yielded improved results.

Regarding the atom types $a$, we compare three approaches for modeling diffusion over these discrete attributes: (**C**) continuous diffusion on one-hot encoded atom types (Jiao et al., 2023), (**C-AB**) continuous diffusion on analog bits (Chen et al., 2023), and (**D**) discrete diffusion with absorbing state (Austin et al., 2021; Shi et al., 2024).

**Metrics** We evaluate generated samples using a machine-learning interatomic potential, based on the open-source pipeline from MATTERGEN (Zeni et al., 2025)—*i.e.* using MatterSim-v1-1M (Yang et al., 2024a). Sample quality is assessed in terms of the following metrics: RMSD between the generated samples and their relaxed structure, where lower values mean generated structures are closer to equilibrium; the average energy above the hull, with lower values meaning that generated materials are closer to thermodynamic (meta-)stability; stability, measured as the proportion of samples with an energy above the convex hull below 0.1 eV/atom; and S.U.N. (stable, unique, novel) measuring the percentage of generated samples that satisfy all three criteria, identifying promising candidates. For each variant of KLDM, we generated 10000 samples, from which we discarded samples that contained elements not supported by the validation pipeline.

**Baselines** We compare KLDM to DIFFCSP and MATTERGEN-MP on MP-20. For the baselines, we report numbers borrowed from MATTERGEN's own benchmark (Zeni et al., 2025). Notably, the baseline models were trained on a re-optimized version of MP-20 in which certain chemical elements were removed, specifically noble gases, radioactive elements, and elements with atomic number greater than 84. Additionally, samples with energy above the hull bigger than 0.1 eV/atom were also filtered out. In contrast, our model was trained on the original, unfiltered MP-20.

**Results** The results are summarized in Table 2. Notably, when relying on analog-bits or discrete diffusion to model the atom types, KLDM outperforms DIFFCSP in terms of RMSD, energy above the hull and stability while being slightly subpar on S.U.N..

Compared to MATTERGEN-MP, KLDM generates structures that are more thermodynamically stable and closer to the convex hull on average, but with slightly higher RMSD, indicating larger deviations from the relaxed geometries. It also produces slightly fewer unique and stable samples, as reflected by a lower S.U.N. score. We attribute this

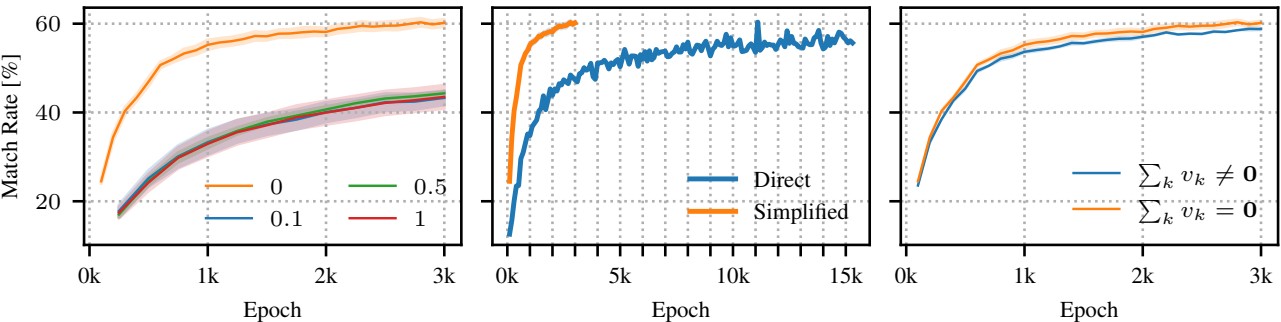

Figure 1: **Ablation study results**. We report the mean and standard error across 6 seeds of the validation Match Rate (MR). (**Left**) Impact of the initial velocity distribution variance. A variance of 0 corresponds to the zero-initial velocity case (i.e., a delta distribution). (**Center**) Effect of the score parameterization on convergence. (**Right**) Impact of enforcing zero-net translation in velocity fields under zero-initial velocities. Notably, all design choices shown contribute to performance improvements.

remaining performance gap to several factors: (1) the more expressive denoiser architecture used by MATTERGEN-MP, which operates in real space, (2) its uses of PC sampler for the lattice parameters, and (3) the impact of dataset pre-processing in the re-optimized MP-20 used for training.

Additional results using standard proxy metrics are provided in Appendix G.

### 5.4. Ablation study

In this section, we analyze the impact of three key design choices: (1) the distribution of the initial velocities, (2) the simplified score parameterization in Eq. (19) compared to the direct parameterization of Eq. (18), and (3) the enforcement of a zero-net translation velocity field.

We find that using zero-initial velocities significantly improves both performance and convergence speed, as shown in Fig. 1 (Left). This improvement may also stem from the ability to use the simplified parameterization from Eq. (19), which is only compatible with zero-initial velocities. As shown in Table 3, this simplified parameterization consistently yields better final performance across different sampling schemes and datasets, while also accelerating convergence (Fig. 1, Center).

Finally, enforcing a zero-net translation velocity field provides additional improvements in terms of MR on the validation set, albeit to a smaller extent. These gains are observed both when using the direct parameterization (Fig. 4) and when combining zero-initial velocities with the simplified parametrization (Fig. 1, Right).

We refer the reader to Appendix E for a more detailed analysis.

### 6. Conclusion

We introduced KLDM, a novel diffusion model for periodic crystal structure generation, whose key innovation lies in the modeling of the fractional coordinates through a coupling with auxiliary variables representing velocities. We evaluated KLDM on both the CSP and DNG tasks, where it performed on par with, or outperformed, several state-of-the-art (SOTA) models. Notably, on the CSP task, KLDM demonstrated significant improvements on the real-scale datasets MPTS-20 and MPTS-52, at a similar computational cost to the compared models. On the DNG task, it matched current SOTA performance as evaluated by machine-learning interatomic potentials. Further validation with DFT simulations is a natural next step to confirm these findings.

Despite its strong performance, there are opportunities for further improvement of KLDM. Exploring alternative processes to noise the velocity variables in Eq. (12) could lead to better results. Additionally, optimizing the score network architecture to better account for the periodic nature of crystal structures (Lin et al., 2023) and investigating modality-specific noise schedules (Qiu et al., 2025) may result in even further gains. We also envision that incorporating space-group information (Jiao et al., 2024) or Wyckoff positions (Levy et al., 2024) could improve practical performance, particularly on the DNG task.

We believe that KLDM provides a strong foundation for diffusion-based crystal structure generation. Future work will focus on targeted material generation with specific properties, as well as extending KLDM to larger systems such as metal-organic frameworks (MOFs) by incorporating rotational frame modeling (Kim et al., 2025).

## Impact statement

Our paper presents a generative model for novel crystalline materials generation. As mentioned in the introduction, the ability to accurately generate novel compounds with targeted properties has significant implications across multiple scientific domains, particularly in molecular discovery and materials design. While advances in materials discovery might benefit society, from more efficient energy storage to improved catalysts, we acknowledge the potential for unintended applications. However, the considerable gap between the model prediction to the successful material synthesis acts as a natural safeguard. Therefore, in the immediate term, our work mostly impacts researchers working on the same topic.

## Acknowledgements

The authors would like to thank the anonymous reviewers for their feedback that helped improve the paper. They would also like to thank Johanna Marie Gegenfurtner for insightful discussions on Lie groups.

FC, AB, and JMGL are supported by the Independent Research Fund Denmark (DELIGHT, Grant No. 0217-00326B; "TeraBatt: Data-driven quest for TWh scalable Na-ion battery", Grant No. 2035-00232B). FB and JF are supported by the Novo Nordisk Foundation through the Center for Basic Machine Learning Research in Life Science (MLLS, grant No. NNF20OC0062606). MNS acknowledges support from the Novo Nordisk Foundation ("BNNs for molecular discovery", Grant No. NNF22OC0076658).

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

## Organization of the Supplementary Material

**Appendix A** We provide a short informal introduction to Lie groups and manifolds. We provide intuition behind key equations Eqs. (12) and (13), and the TDM and KLDM models more broadly.

**Appendix B** We discuss why data on a torus can be represented either as angles or rotation matrices. We also show that, in this case, the matrix exponential simplifies to a rotation matrix, enabling a coordinate update that corresponds to a translation following by a wrapping operation.

**Appendix C** We derive, step-by-step, the target used to train the score network in KLDM.

**Appendix D** We analyze how enforcing a zero-net translation constraint affects the mean of the system.

**Appendix E** We provide an additional discussion and analysis of the ablation study.

**Appendix F and Appendix G** We present additional results for the CSP and DNG tasks, respectively.

**Appendix H** We present pseudo-code for all the core algorithms in KLDM.

**Appendix I** Experimental settings required for reproducibility, along with a brief discussion of the differences between KLDM and the baseline methods.

## A. A primer on Lie groups and manifolds

In this section, we provide an intuitive overview of key concepts used in the main text. For crystalline materials, the fractional coordinates $\boldsymbol{f}$, which define atomic positions within the unit cell, lie on a hypertorus: *i.e.* $\boldsymbol{f} \in [0,1)^{3 \times K} \cong \mathbb{T}^{3 \times K}$. The hypertorus is a direct product of Lie groups. In the main paper, we introduced Lie groups as smooth manifolds equipped with a group structure $G$ and smooth group operations. Here, we revisit some of the building concepts underlying KLDM in more detail, from a Riemannian geometry perspective rather than from the Lie group one.

**Lie group** In mathematics, a *group* is denoted by $(G, \cdot)$, with $G$ being the non-empty set of elements that belong to the group and $\cdot$ being the group operation that combines any two elements $a, b$ in the group $G$ and results in an element of $G$. The group operation $\cdot$ has to satisfy three different properties:

- $(a \cdot b) \cdot c = a \cdot (b \cdot c) \quad \forall a, b, c \in G$ (*associativity* property)

- For every element $a \in G$, there should be an unique element $e \in G$, such that $e \cdot a = a$ and $a \cdot e = a$ (existence of an unique *identity element*)

- For every element $a \in G$ there is an element $b \in G$ such that $a \cdot b = e$ and $b \cdot a = e$. (existence of an *inverse element*)

If the group operation $\cdot$ is also *commutative*, i.e. $a \cdot b = b \cdot a \quad \forall a, b \in G$, then the group is called an Abelian group. This property holds in the case of the hypertorus $\mathbb{T}^{3 \times K}$ and allows us to define the transition kernel in Eq. (16) in closed-form (and hence a simulation-free training procedure). For a complete derivation of this, we refer to Zhu et al. (2024).

We can formalize two different operations that can be done with elements of the group $G$ that will be useful to give some intuition about why data on a manifold can be modeled through quantities defined in Euclidean space. Let us consider two elements $a, g \in G$ of the group. The *left translation* of $g$ by $a$ is defined as the operation $L_a : G \to G$ which takes $g$ as input and returns $a \cdot g$, or equivalently $g \mapsto a \cdot g$. In the same way, we can define the right translation as $R_a : G \to G, g \mapsto g \cdot a$.

**Tangent space** Before describing other key elements of a Lie group, we introduce the concept of *tangent space*, which stems from the fact that a Lie group is also a manifold. Possible ways to understand the notion of a manifold involve thinking about it as a hypersurface embedded in a higher-dimensional space or as a collection of points that are somewhat connected to generate a surface. A tangent space is a vector space containing all the vectors that are tangential to a specific point in the manifold, *i.e.* any element $g \in G$ in the case of a Lie group. It is usually denoted as $\mathcal{T}_g \mathcal{M}$ for manifolds and $\mathcal{T}_g G$ for Lie Groups. The tangent space offers a linearized view of the manifold in the neighborhood of the point $g$ and therefore it is usually considered as Euclidean space—*i.e.* $\mathcal{T}_g G \cong \mathbb{R}^d$, with $d$ being the dimension of $g$. If we consider all the elements in the group $G$, the set of all the tangent spaces associated with these elements form the *tangent bundle*, denoted with $\mathcal{T}\mathcal{M}$ or $\mathcal{T}G$, depending if we are working with a manifold or a Lie group.

**Lie algebra, left-invariant vector fields and exponential maps** The Lie algebra $\mathfrak{g}$ is defined as the tangent space at the identity element of the group, $e \in G$: $\mathcal{T}_e G$. Since tangent spaces are approximately Euclidean, we treat the Lie algebra $\mathfrak{g}$ as Euclidean as well.

To understand how to define the dynamics in Eq. (12) and introduced by Zhu et al. (2024), we first introduce *left-invariant vector fields*.

A vector field, $X : G \to \mathcal{T}G$, is a function that associates every element of the group $g \in G$ to a vector $X_g \in \mathcal{T}_g G$ of the tangent space at $g$.

Recall the left-translation operation $L_a : G \to G$, which maps a group element $g \in G$ to $a \cdot g \in G$ for any group element $a \in G$. The differential of this operation, denoted as $(dL_a)_g : \mathcal{T}_g G \to \mathcal{T}_{a \cdot g} G$, acts on tangent vectors instead of group elements. A vector field is *left-invariant* if:

$$(dL_a)_g(X_g) = X_{a \cdot g}, \quad \text{for any two group elements } a, g \in G. \tag{20}$$

Intuitively, this means that the vector field behaves predictably under left-translations. Specifically, if $g$ is translated to $a \cdot g$, the vector $X_g \in \mathcal{T}_g G$ will be mapped to $X_{a \cdot g}$. Thus, a left-invariant vector looks the same at all points in the group.

In particular, it can be shown that there exists a linear isomorphism between the Lie algebra $\mathfrak{g}$ and the tangent space of the identity element $e \in G$, *i.e.* $\mathfrak{g} \cong \mathcal{T}_e G$ (Arvanitogeōrgos, 2003). This means that an invariant vector field is determined entirely by its value on the tangent space of the identity element $e \in G$. Using this, we can rewrite Eq. (20) in terms of the identity element $e \in G$ and its tangent space $\mathcal{T}_e G$, which is $\mathfrak{g}$:

$$
\begin{aligned}
(dL_a)_e &: \mathcal{T}_e G \to \mathcal{T}_{a \cdot e} G = \mathcal{T}_a G \\
(dL_a)_e(X_e) &= X_{a \cdot e} = X_a
\end{aligned}
\tag{21}
$$

where $a \cdot e = a$. Eq. (21) shows that any vector $X_e \in \mathcal{T}_e G$ can be translated to a tangent space $X_a \in \mathcal{T}_a G$ using $(dL_a)_e(X_e)$. Now, with both the group element $a \in G$ and the tangent vector $X_a \in \mathcal{T}_a G$, we can use the *Riemannian exponential map*[4] $\exp_a : \mathcal{T}_a G \to G$ to compute the group element $a' \in G$ reached by starting from $a$ with velocity $X_a$ in one unit time. This map extends the Euclidean notion of motion along a straight line to the manifold.

**How does this relate to modeling data living on the torus?** We can take advantage of the torus being a Lie group and use the operations from the previous section. Specifically, we use the Lie algebra $\mathfrak{g}$ (a Euclidean vector space) and the operation mapping vectors in $\mathfrak{g}$ to the tangent space of any point of the manifold (via Eq. (21)). In the forward dynamics of KLDM (Eq. (12)), we couple fractional coordinates (elements of our Lie group) with velocity vectors in $\mathfrak{g}$. Since the torus is a Lie group, we can map any vector in $\mathfrak{g}$ the tangent space of any group element. This allows us to define a standard Euclidean diffusion model over these velocities in $\mathfrak{g}$ and use Eq. (20) to get the corresponding tangential velocities. By applying the exponential map, we can update the fractional coordinates, explaining the coupled process in Eq. (12).

The computation of Eq. (21) for the torus is straightforward, as both the vectors in the Lie algebra $\mathfrak{g}$ and group elements can be represented as $d \times d$ matrices. The differential operation in Eq. (21) simplifies to matrix multiplication (Zhu et al., 2024). Given a vector $X_e \in \mathfrak{g}$ and an element $a \in G$, the tangent vector in $a$ is simply $X_a = a X_e \in T_a G$.

Additionally, the Riemannian exponential map, involving constant velocity, has a closed-form solution using the matrix exponential. The group element $a'$ reached from $a$ with velocity $X_a$ is given by $a' = \exp_a(a X_e) = \exp_a(X_a) = a \operatorname{expm}(X_e)$, where $\operatorname{expm}(X_e)$ denotes the matrix exponential.

## B. Sampling from the noising and denoising processes

### B.1. Intuition about the correspondence between $\mathbb{T}$ and $\mathrm{SO}(2)$

In the following, we provide an intuitive explanation of the isomorphism between $\mathbb{T}$ and $\mathrm{SO}(2)$. Let us consider a variable $x \in [a, b)$, we are going to work on an *alternative representation* thereof, that we will call $g$ – a representation in $\mathrm{SO}(2)$.

---

[4]Note that the exponential map operation can be defined from a Riemannian manifold perspective and from the Lie group one. In the Riemannian case, we have that it is defined as $\exp_a : \mathcal{T}_a G \to G$, i.e. it takes a velocity living in the tangent space of an element of the manifold (which in this case is also a group) and map it to another element of the manifold. The Lie group exponential map instead is a function $\exp_e : \mathfrak{g} \to G$ that takes a vector in the Lie algebra $\mathfrak{g} \cong \mathcal{T}_e G$ and maps it to the group.

First, we are going to map $x$ to an angle $\theta$ as follows,

$$\theta = 2\pi \left( \frac{x}{b-a} - \frac{1}{2} \right),$$

such that $\theta \in [-\pi, \pi)$.

Then, we can construct the representation $g$ as,

$$g = \begin{bmatrix} \cos\theta & \sin\theta \\ -\sin\theta & \cos\theta \end{bmatrix}.$$

From $g$, we can readily recover $\theta$,

$$\theta = \text{sign}(g_{0,1}) \cdot \arccos g_{0,0},$$

and in turn $x$,

$$x = \left( \frac{\theta}{2\pi} + \frac{1}{2} \right) (b-a).$$

## B.2. Update of positions is equivalent to periodic translation

As we have mentioned in the previous section, the update of the coordinate in the forward process can be expressed as $\boldsymbol{f}_t = \boldsymbol{f}_0 \, \text{expm}(\boldsymbol{r}_t)$ where then $\boldsymbol{r}_t \sim \text{WN}(\boldsymbol{r}_t | \mu_{\boldsymbol{r}_t}, \sigma^2_{\boldsymbol{r}_t})$. Therefore, the update involves a matrix exponential operation. In the case of the torus, we have that the structure of $\boldsymbol{r}_t$ can be written as a $2 \times 2$ skew-symmetric matrix of the form:

$$\boldsymbol{r}_t = \begin{bmatrix} 0 & r_t \\ -r_t & 0 \end{bmatrix}, \qquad r_t \in \mathbb{R}$$

In our case, we are interested in computing the matrix exponential of $\boldsymbol{r}_t$. For simplicity, if we assume that $r_t = 1$, we are interested in computing a matrix exponential with the following structure:

$$\text{expm}(At) = \text{expm}\left( \begin{bmatrix} 0 & 1 \\ -1 & 0 \end{bmatrix} t \right)$$

This corresponds to the following infinite sum

$$\text{expm}(At) = \text{expm}\left( \begin{bmatrix} 0 & 1 \\ -1 & 0 \end{bmatrix} t \right) = \begin{bmatrix} 1 & 0 \\ 0 & 1 \end{bmatrix} + \begin{bmatrix} 0 & t \\ -t & 0 \end{bmatrix} + \frac{1}{2} \begin{bmatrix} -t^2 & 0 \\ 0 & -t^2 \end{bmatrix} + \frac{1}{3!} \begin{bmatrix} 0 & -t^3 \\ t^3 & 0 \end{bmatrix} + \frac{1}{4!} \begin{bmatrix} t^4 & 0 \\ 0 & t^4 \end{bmatrix} + \cdots$$

which can be rewritten as follows:

$$\text{expm}\left( \begin{bmatrix} 0 & 1 \\ -1 & 0 \end{bmatrix} t \right) = \begin{bmatrix} 1 - \frac{t^2}{2} + \frac{t^4}{4!} - \cdots & t - \frac{t^3}{3!} + \frac{t^5}{5!} - \cdots \\ -t + \frac{t^3}{3!} - \frac{t^5}{5!} + \cdots & 1 - \frac{t^2}{2} + \frac{t^4}{4!} - \cdots \end{bmatrix}$$

where we can recognize the Maclaurin series for $\sin(t)$ and $\cos(t)$. Therefore, we can conclude that:

$$\text{expm}\left( \begin{bmatrix} 0 & 1 \\ -1 & 0 \end{bmatrix} t \right) = \begin{bmatrix} \cos(t) & \sin(t) \\ -\sin(t) & \cos(t) \end{bmatrix}$$

In forward and backward integration of the dynamics described by Eq. (7) and Eq. (13) the update of the coordinates involves the computation of a matrix exponential. In the following, we are going to derive step-by-step the closed-form solution for the matrix exponential, highlighting how this will be equivalent to a translation and a wrap of the current position.

**Forward process** In the forward process, assuming a discretization step denoted by $\mathrm{d}t$, we are interested in computing the following update:

$$
\begin{aligned}
\boldsymbol{f}_t &= \boldsymbol{f}_{t-1}\mathrm{expm}(\mathrm{d}t\boldsymbol{v}_t) \\
&= \boldsymbol{f}_{t-1}\,\mathrm{expm}\left(\mathrm{d}t\begin{bmatrix} 0 & v_t \\ -v_t & 0 \end{bmatrix}\right) = \boldsymbol{f}_{t-1}\,\mathrm{expm}\left(\begin{bmatrix} 0 & v_t\mathrm{d}t \\ -v_t\mathrm{d}t & 0 \end{bmatrix}\right) \\
&= \boldsymbol{f}_{t-1}\begin{bmatrix} \cos v_t\mathrm{d}t & \sin v_t\mathrm{d}t \\ -\sin v_t\mathrm{d}t & \cos v_t\mathrm{d}t \end{bmatrix} = \begin{bmatrix} \cos\theta & \sin\theta \\ -\sin\theta & \cos\theta \end{bmatrix}\begin{bmatrix} \cos v_t\mathrm{d}t & \sin v_t\mathrm{d}t \\ -\sin v_t\mathrm{d}t & \cos v_t\mathrm{d}t \end{bmatrix} \\
&= \begin{bmatrix} \cos\theta\cos v_t\mathrm{d}t - \sin\theta\sin v_t\mathrm{d}t & \cos\theta\sin v_t\mathrm{d}t + \sin\theta\cos v_t\mathrm{d}t \\ -\sin v_t\mathrm{d}t\cos\theta - \cos v_t\mathrm{d}t\sin\theta & -\sin\theta\sin v_t\mathrm{d}t + \cos\theta\cos v_t\mathrm{d}t \end{bmatrix} \\
&= \begin{bmatrix} \cos(\theta + v_t\mathrm{d}t) & \sin(\theta + v_t\mathrm{d}t) \\ -\sin(\theta + v_t\mathrm{d}t) & \cos(\theta + v_t\mathrm{d}t) \end{bmatrix}
\end{aligned}
$$

where we used the rotation matrix representation for the fractional coordinates on the torus, as we explained at the beginning of this section.

**Reverse process** In the backward process, instead, we are interested in computing the following update:

$$
\begin{aligned}
\boldsymbol{f}_t &= \boldsymbol{f}_{t-1}\mathrm{expm}(-\mathrm{d}t\boldsymbol{v}_t) \\
&= \boldsymbol{f}_{t-1}\,\mathrm{expm}\left(-\mathrm{d}t\begin{bmatrix} 0 & v_t \\ -v_t & 0 \end{bmatrix}\right) = \boldsymbol{f}_{t-1}\,\mathrm{expm}\left(\begin{bmatrix} 0 & -v_t\mathrm{d}t \\ v_t\mathrm{d}t & 0 \end{bmatrix}\right) \\
&= \boldsymbol{f}_{t-1}\begin{bmatrix} \cos v_t\mathrm{d}t & -\sin v_t\mathrm{d}t \\ \sin v_t\mathrm{d}t & \cos v_t\mathrm{d}t \end{bmatrix} = \begin{bmatrix} \cos\theta & \sin\theta \\ -\sin\theta & \cos\theta \end{bmatrix}\begin{bmatrix} \cos v_t\mathrm{d}t & -\sin v_t\mathrm{d}t \\ \sin v_t\mathrm{d}t & \cos v_t\mathrm{d}t \end{bmatrix} \\
&= \begin{bmatrix} \cos\theta\cos v_t\mathrm{d}t + \sin\theta\sin v_t\mathrm{d}t & -\cos\theta\sin v_t\mathrm{d}t + \sin\theta\cos v_t\mathrm{d}t \\ -\cos v_t\mathrm{d}t\sin\theta + \sin v_t\mathrm{d}t\cos\theta & \sin\theta\sin v_t\mathrm{d}t + \cos\theta\cos v_t\mathrm{d}t \end{bmatrix} \\
&= \begin{bmatrix} \cos(\theta - v_t\mathrm{d}t) & \sin(\theta - v_t\mathrm{d}t) \\ -\sin(\theta - v_t\mathrm{d}t) & \cos(\theta - v_t\mathrm{d}t) \end{bmatrix}
\end{aligned}
$$

## C. Derivation of the target of KLDM

We recall that the transition kernel of our KLDM forward process defined in Eq. (12) can be obtained in closed form and it is given by (Zhu et al., 2024):

$$
p_{t|0}(\boldsymbol{f}_t, \boldsymbol{v}_t | \boldsymbol{f}_0, \boldsymbol{v}_0) = \mathrm{WN}\left(\mathrm{logm}(\boldsymbol{f}_0^{-1}\boldsymbol{f}_t) | \boldsymbol{\mu}_{\boldsymbol{r}_t}, \sigma_{\boldsymbol{r}_t}^2\mathbf{I}\right) \cdot \mathcal{N}_{\boldsymbol{v}}\left(\boldsymbol{v}_t | \boldsymbol{\mu}_{\boldsymbol{v}_t}, \sigma_{\boldsymbol{v}_t}^2\mathbf{I}\right),
$$

where the mean and the variance of the two distributions are the following:

$$
\boldsymbol{\mu}_{\boldsymbol{r}_t} = \frac{1 - e^{-t}}{1 + e^{-t}}(\boldsymbol{v}_t + \boldsymbol{v}_0) \qquad \boldsymbol{\mu}_{\boldsymbol{v}_t} = e^{-t}\boldsymbol{v}_0 \tag{22}
$$

$$
\sigma_{\boldsymbol{r}_t}^2 = 2t + \frac{8}{e^t + 1} - 4 \qquad \sigma_{\boldsymbol{v}_t}^2 = 1 - e^{-2t} \tag{23}
$$

Intuitively, the normal distribution describes how we should noise the initial velocity $\boldsymbol{v}_0$ to get a sample $\boldsymbol{v}_t$, while the wrapped-normal distribution implicitly defines how to get a noisy sample for the fractional coordinates $\boldsymbol{f}_t$ starting from $\boldsymbol{f}_0$. The trick is to define $\boldsymbol{r}_t = \mathrm{logm}(\boldsymbol{f}_0^{-1}\boldsymbol{f}_t)$, and by taking the matrix-exponential on both sides, we can get the following update rule $\boldsymbol{f}_t = \boldsymbol{f}_0\,\mathrm{expm}(\boldsymbol{r}_t)$ where then $\boldsymbol{r}_t \sim \mathrm{WN}\left(\boldsymbol{r}_t | \boldsymbol{\mu}_{\boldsymbol{r}_t}, \sigma_{\boldsymbol{r}_t}^2\mathbf{I}\right)$

The target score used in the DSM loss (Eq. (9)) for KLDM is $\nabla_{\boldsymbol{v}_t}\log p_{t|0}(\boldsymbol{f}_t, \boldsymbol{v}_t | \boldsymbol{f}_0, \boldsymbol{v}_0)$. In the following, we derive it step-by-step, highlighting at the end the parameterization used to train our *score network*. The target score can therefore be

computed as follows:

$$\nabla_{\boldsymbol{v}_t} \log p_{t|0}(\boldsymbol{f}_t, \boldsymbol{v}_t | \boldsymbol{f}_0, \boldsymbol{v}_0) = \nabla_{\boldsymbol{v}_t} \log \left[ \text{WN}\big( \text{logm}(\boldsymbol{f}_0^{-1} \boldsymbol{f}_t) | \boldsymbol{\mu}_{\boldsymbol{r}_t}, \sigma_{\boldsymbol{r}_t}^2 \mathbf{I} \big) \cdot \mathcal{N}_{\boldsymbol{v}}\big( \boldsymbol{v}_t | \boldsymbol{\mu}_{\boldsymbol{v}}, \sigma_{\boldsymbol{v}}^2 \mathbf{I} \big) \right]$$

$$= \underbrace{\nabla_{\boldsymbol{v}_t} \log \text{WN}\big( \text{logm}(\boldsymbol{f}_0^{-1} \boldsymbol{f}_t) | \boldsymbol{\mu}_{\boldsymbol{r}_t}, \sigma_{\boldsymbol{r}_t}^2 \mathbf{I} \big)}_{\boldsymbol{s}_{\boldsymbol{c}} \text{ coordinates term}} + \underbrace{\nabla_{\boldsymbol{v}_t} \log \mathcal{N}_{\boldsymbol{v}}\big( \boldsymbol{v}_t | \boldsymbol{\mu}_{\boldsymbol{v}}, \sigma_{\boldsymbol{v}}^2 \mathbf{I} \big)}_{\boldsymbol{s}_{\boldsymbol{v}} \text{ velocity term}} \quad (24)$$

We start by deriving the score for the second term $\boldsymbol{s}_{\boldsymbol{v}}$, which can be expressed in multiple way:

$$\boldsymbol{s}_{\boldsymbol{v}} = \frac{-\boldsymbol{v}_t + \boldsymbol{\mu}_{\boldsymbol{v}}}{\sigma_{\boldsymbol{v}}^2} = -\frac{\varepsilon}{\sigma_{\boldsymbol{v}}} \quad (25)$$

We can now focus on the $\boldsymbol{s}_{\boldsymbol{c}}$ term, where for simplicity we define $\boldsymbol{r}_t = \text{logm}(\boldsymbol{f}_0^{-1} \boldsymbol{f}_t)$ as above:

$$\boldsymbol{s}_{\boldsymbol{c}} = \nabla_{\boldsymbol{v}_t} \log \text{WN}\big( \text{logm}(\boldsymbol{f}_0^{-1} \boldsymbol{f}_t) | \boldsymbol{\mu}_{\boldsymbol{r}_t}, \sigma_{\boldsymbol{r}_t}^2 \mathbf{I} \big) = \nabla_{\boldsymbol{v}_t} \log \text{WN}\big( \boldsymbol{r}_t | \boldsymbol{\mu}_{\boldsymbol{r}_t}, \sigma_{\boldsymbol{r}_t}^2 \mathbf{I} \big)$$

$$= \nabla_{\boldsymbol{v}_t} \log \left( \frac{1}{\sqrt{2\pi}\sigma_{\boldsymbol{r}_t}} \sum_{k=-\infty}^{+\infty} \exp\left( -\frac{(\boldsymbol{r}_t - \boldsymbol{\mu}_{\boldsymbol{r}_t} + 2\pi k)^2}{2\sigma_{\boldsymbol{r}_t}^2} \right) \right)$$

$$= \nabla_{\boldsymbol{v}_t} \log \left( \sum_{k=-\infty}^{+\infty} \exp\left( -\frac{(\boldsymbol{r}_t - \boldsymbol{\mu}_{\boldsymbol{r}_t} + 2\pi k)^2}{2\sigma_{\boldsymbol{r}_t}^2} \right) \right)$$

$$= \underbrace{\frac{1}{\sum_{k=-\infty}^{+\infty} \exp\left( -\frac{(\boldsymbol{r}_t - \boldsymbol{\mu}_{\boldsymbol{r}_t} + 2\pi k)^2}{2\sigma_{\boldsymbol{r}_t}^2} \right)}}_{C} \nabla_{\boldsymbol{v}_t} \left[ \sum_{k=-\infty}^{+\infty} \exp\left( -\frac{(\boldsymbol{r}_t - \boldsymbol{\mu}_{\boldsymbol{r}_t} + 2\pi k)^2}{2\sigma_{\boldsymbol{r}_t}^2} \right) \right]$$

$$= C \cdot \left[ \sum_{k=-\infty}^{+\infty} \nabla_{\boldsymbol{v}_t} \exp\left( -\frac{(\boldsymbol{r}_t - \boldsymbol{\mu}_{\boldsymbol{r}_t} + 2\pi k)^2}{2\sigma_{\boldsymbol{r}_t}^2} \right) \right]$$

$$= C \cdot \left[ \sum_{k=-\infty}^{+\infty} \exp\left( -\frac{(\boldsymbol{r}_t - \boldsymbol{\mu}_{\boldsymbol{r}_t} + 2\pi k)^2}{2\sigma_{\boldsymbol{r}_t}^2} \right) \nabla_{\boldsymbol{v}_t} \left( -\frac{(\boldsymbol{r}_t - \boldsymbol{\mu}_{\boldsymbol{r}_t} + 2\pi k)^2}{2\sigma_{\boldsymbol{r}_t}^2} \right) \right]$$

$$= C \cdot \left[ \sum_{k=-\infty}^{+\infty} \exp\left( -\frac{(\boldsymbol{r}_t - \boldsymbol{\mu}_{\boldsymbol{r}_t} + 2\pi k)^2}{2\sigma_{\boldsymbol{r}_t}^2} \right) \nabla_{\boldsymbol{v}_t} \left( -\frac{(\boldsymbol{r}_t - \frac{1-e^{-t}}{1+e^{-t}}(\boldsymbol{v}_t + \boldsymbol{v}_0) + 2\pi k)^2}{2\sigma_{\boldsymbol{r}_t}^2} \right) \right]$$

$$= C \cdot \left[ \sum_{k=-\infty}^{+\infty} \exp\left( -\frac{(\boldsymbol{r}_t - \boldsymbol{\mu}_{\boldsymbol{r}_t} + 2\pi k)^2}{2\sigma_{\boldsymbol{r}_t}^2} \right) \frac{1-e^{-t}}{1+e^{-t}} \frac{1}{\sigma_{\boldsymbol{r}_t}^2} \left( \boldsymbol{r}_t - \frac{1-e^{-t}}{1+e^{-t}}(\boldsymbol{v}_t + \boldsymbol{v}_0) + 2\pi k \right) \right]$$

$$= \frac{1}{\sum_{k=-\infty}^{+\infty} \exp\left( -\frac{(\boldsymbol{r}_t - \boldsymbol{\mu}_{\boldsymbol{r}_t} + 2\pi k)^2}{2\sigma_{\boldsymbol{r}_t}^2} \right)} \left[ \sum_{k=-\infty}^{+\infty} \frac{1-e^{-t}}{1+e^{-t}} \frac{1}{\sigma_{\boldsymbol{r}_t}^2} \left( \boldsymbol{r}_t - \frac{1-e^{-t}}{1+e^{-t}}(\boldsymbol{v}_t + \boldsymbol{v}_0) + 2\pi k \right) \exp\left( -\frac{(\boldsymbol{r}_t - \boldsymbol{\mu}_{\boldsymbol{r}_t} + 2\pi k)^2}{2\sigma_{\boldsymbol{r}_t}^2} \right) \right]$$

$$= \frac{1}{\sum_{k=-\infty}^{+\infty} \exp\left( -\frac{(\boldsymbol{r}_t - \boldsymbol{\mu}_{\boldsymbol{r}_t} + 2\pi k)^2}{2\sigma_{\boldsymbol{r}_t}^2} \right)} \left[ \sum_{k=-\infty}^{+\infty} \frac{1-e^{-t}}{1+e^{-t}} \frac{1}{\sigma_{\boldsymbol{r}_t}^2} \left( \boldsymbol{r}_t - \boldsymbol{\mu}_{\boldsymbol{r}_t} + 2\pi k \right) \exp\left( -\frac{(\boldsymbol{r}_t - \boldsymbol{\mu}_{\boldsymbol{r}_t} + 2\pi k)^2}{2\sigma_{\boldsymbol{r}_t}^2} \right) \right]$$

$$= \frac{1-e^{-t}}{1+e^{-t}} \frac{1}{\sum_{k=-\infty}^{+\infty} \exp\left( -\frac{(\boldsymbol{r}_t - \boldsymbol{\mu}_{\boldsymbol{r}_t} + 2\pi k)^2}{2\sigma_{\boldsymbol{r}_t}^2} \right)} \left[ \sum_{k=-\infty}^{+\infty} \frac{1}{\sigma_{\boldsymbol{r}_t}^2} \left( \boldsymbol{r}_t - \boldsymbol{\mu}_{\boldsymbol{r}_t} + 2\pi k \right) \exp\left( -\frac{(\boldsymbol{r}_t - \boldsymbol{\mu}_{\boldsymbol{r}_t} + 2\pi k)^2}{2\sigma_{\boldsymbol{r}_t}^2} \right) \right]$$

$$= \frac{1-e^{-t}}{1+e^{-t}} \nabla_{\boldsymbol{\mu}_{\boldsymbol{r}_t}} \log \text{WN}\big( \boldsymbol{r}_t | \boldsymbol{\mu}_{\boldsymbol{r}_t}, \sigma_{\boldsymbol{r}_t}^2 \mathbf{I} \big)$$

$$= \frac{\partial \boldsymbol{\mu}_{\boldsymbol{r}_t}}{\partial \boldsymbol{v}_t} \nabla_{\boldsymbol{\mu}_{\boldsymbol{r}_t}} \log \text{WN}\big( \boldsymbol{r}_t | \boldsymbol{\mu}_{\boldsymbol{r}_t}, \sigma_{\boldsymbol{r}_t}^2 \mathbf{I} \big) \quad (26)$$

Therefore, the target score is given by summing Eq. (25) and Eq. (26) together. If we now assume that the velocities are

zeros at time $t = 0$, we can notice that the score $\boldsymbol{s_v}$ can be rewritten as follows:

$$\boldsymbol{s_v} = \frac{-\boldsymbol{v}_t + \boldsymbol{\mu_v}}{\sigma_v^2} = \frac{-\boldsymbol{v}_t + e^{-t}\boldsymbol{v}_0}{\sigma_v^2} = -\frac{\boldsymbol{v}_t}{\sigma_v^2} \tag{27}$$

where we used the definition of $\boldsymbol{\mu_v} = e^{-t}\boldsymbol{v}_0$ and the fact that we assumed that $\boldsymbol{v}_0 = \boldsymbol{0}$, resulting in $\boldsymbol{\mu_v} = \boldsymbol{0}$.

## D. Effect of the zero-net translation velocity field

**Mismatch between translation-invariant score parameterization and non-invariant training target**   As noted by Lin et al. (2024), there is a potential mismatch between the translation-invariant parameterization of the score and the non-invariant training target. For instance, a noisy point cloud and its periodic translation are considered equivalent by the network, but the target scores, as computed in Eq. (18) and used by DIFFCSP (Jiao et al., 2023), will differ. While this mismatch is averaged out during training and does not prevent the model from learning a useful score (as DIFFCSP performs well on the tasks), it still affects gradient variance during training.

**Constraining the velocity field**   In Section 3.2, we enforce a velocity field with zero net translation by imposing zero-mean velocities ($\sum_k^K \boldsymbol{v}_k = \boldsymbol{0}$). Since our score network is periodic translation-invariant, the goal is to remove any translation component from the dynamics, as the network cannot distinguish between translations.

To illustrate this, consider the simple case of a one-dimensional system with a single atom (*i.e.* $\boldsymbol{f}_0$ is just one coordinate). In this scenario, any noisy $\boldsymbol{f}_t$ can be seen as a periodic translation of any other $\boldsymbol{f}'_t$, and hence even of the clean sample $\boldsymbol{f}_0$ itself. Without constraining the velocity field, the dynamics defined in Eq. (12) produce noisy samples $\boldsymbol{f}_t$ corresponding to periodic translations of $\boldsymbol{f}_0$, which the network cannot distinguish. By imposing a zero-mean velocity field, the velocity coupled with $\boldsymbol{f}_0$ must, in this case, be zero, preventing the forward dynamics from introducing an overall translation of the crystal.

**Effect of the constraint on forward dynamics**   To understand the impact of this constraint on the forward dynamics in the general case, we track the *mean* of the noisy samples generated by using the transition kernel defined in Eq. (16). Since our data resides on a torus, the appropriate generalization of the concept of arithmetic mean to non-Euclidean spaces is the Fréchet mean (Fréchet, 1948). The Fréchet mean ensures that the resulting point lies on the manifold but requires solving an optimization problem. Given $N$ datapoints $\{\boldsymbol{f}_0, \cdots, \boldsymbol{f}_N\} \in \mathcal{M}$ on a manifold, we define the *Karcher means* as the set of solutions to:

$$\boldsymbol{f}_{\text{mean}} = \arg\min_{\boldsymbol{p}\in\mathcal{M}} \sum_{i=1}^{N} d^2(\boldsymbol{p}, \boldsymbol{f}_i), \tag{28}$$

where $d(\cdot,\cdot)$ is the geodesic distance on the manifold. When the optimization yields a unique solution, it is the Fréchet mean.

We consider a one-dimensional crystal composed of 10 atoms and two different noise levels, namely $\sigma^2 = 0.1$ and $\sigma^2 = 0.7$. We sample 1000 points from the transition kernel defined in Eq. (16). For each sample, we check whether the original Fréchet mean is preserved when enforcing a zero-net velocity field. As shown in Fig. 2, at low noise levels, the Fréchet mean is generally preserved, indicating that explicitly ensuring a zero-net velocity field prevents the translation of the whole system. However, at higher levels of noise, we note that the Fréchet mean jumps between $n$ different values, each of them separated by $\frac{2\pi}{n}$. Despite this, the noisy samples typically share the same mean as the noiseless reference. In contrast, when no constraint is imposed on the velocity field (Fig. 3), the Fréchet mean is never preserved for both noise levels, and it varies continuously. Our approach, which constrains the velocity field, limits the Fréchet mean to a discrete set of possible values, as opposed to the continuous variation observed without the constraint.

The above analysis shows that sampling from the transition kernel during training can introduce translations, meaning the mismatch may not be fully mitigated across all noise levels. However, we argue that this mismatch is significantly reduced at low noise levels when the vector field constraint is enforced, which benefits training (as shown in Appendix E). Enforcing the constraint also turns the issue of shifting the mean of the system into a discrete problem, rather than a continuous one.

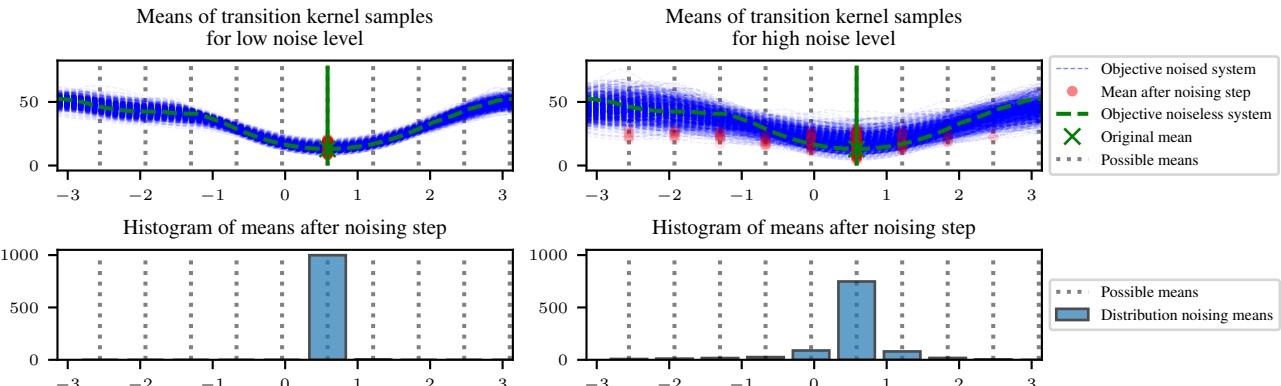

Figure 2: Effect of sampling from the transition kernel in Eq. (16) *with a zero-net translation velocity field.*. We consider a 1D crystal made of 10 atoms and two different noise levels: low noise (sampled from $\mathcal{N}(0, 0.1)$, *left*) and high noise (sampled from $\mathcal{N}(0, 0.7)$, *right*). For each noise level, we generate 1000 noisy samples. At low noise, enforcing a zero-net translation velocity field preserves the original mean, preventing any system translation. At higher noise levels, while the mean is typically conserved, it occasionally shifts between 10 distinct values, each separated by $\frac{2\pi}{10}$.

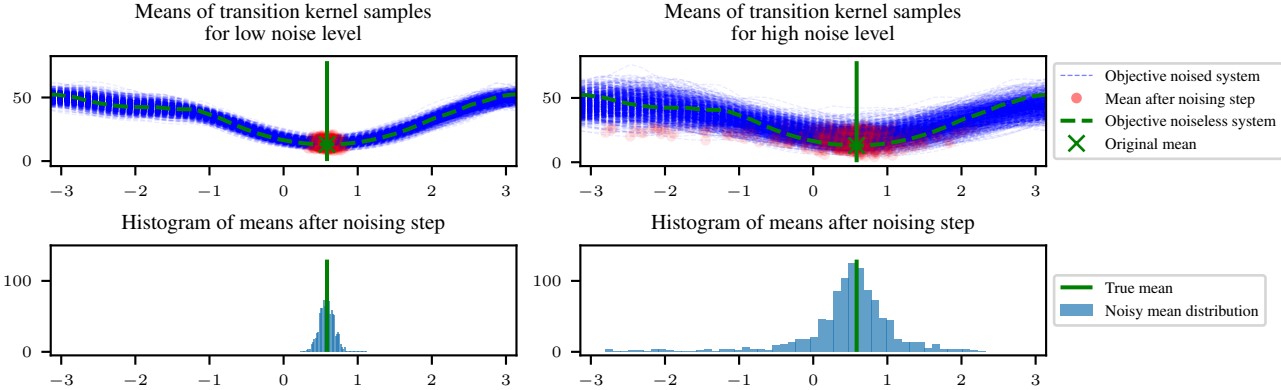

Figure 3: Effect of sampling from the transition kernel in Eq. (16) *without a zero-net translation velocity field.*. We consider a 1D crystal made of 10 atoms and two different noise levels: low noise (sampled from $\mathcal{N}(0, 0.1)$, *left*) and high noise (sampled from $\mathcal{N}(0, 0.7)$, *right*). For each noise level, we generate 1000 noisy samples. We observe that the mean is not preserved at either low or high noise levels, as it changes continuously.

# E. Ablation study

In this section, we analyze the effect of the different design choices of our method. We focus on the initial velocity distributions, the score parameterization, and the translation-free velocity fields.

**Initial-velocity distributions** KLDM uses zero-initial velocities, which intuitively corresponds to considering the coordinates $f$ at time $t = 0$ as being at rest, allowing the noise to gradually propagate from the velocity to these variables. This also allows us to use the simplified parameterization for the score formulated in Eq. (19) ablated above. In Fig. 1 *(Left)*, we compare the choice of using zero initial velocities against having a zero-mean Gaussian distribution with three different variance values. We note there is a strong benefit in using zero initial velocities, potentially explained by the simplified parameterization.

**Velocity field** The last design choice we consider is the enforcement of a velocity field with a zero-net translation. We analyze the effect of these choices both for zero initial velocities and therefore simplified score parameterization, and in the case of non-zero initial velocities and direct parameterization. We report results in Fig. 1 *(Right)*. By removing the degree of freedom of modeling overall translations of the system, we observe a gain, albeit marginal, in terms of validation set match rate during training in all cases, in particular with non-zero initial velocities, as displayed in Fig. 4.

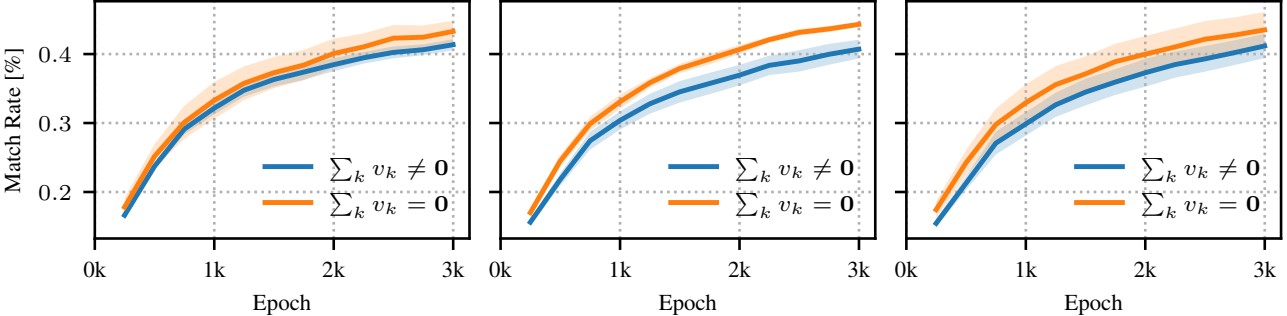

Figure 4: Ablation of velocity fields with zero net translation for non-zero initial velocities. (**Left**) Initial velocities are sampled from a $\mathcal{N}(\mathbf{0}, 0.1 \cdot \mathbf{I})$, i.e. $\sigma_{\boldsymbol{v}_0}^2 = 0.1$. (**Center**) Initial velocities are sampled from a $\mathcal{N}(\mathbf{0}, 0.5 \cdot \mathbf{I})$, i.e. $\sigma_{\boldsymbol{v}_0}^2 = 0.5$ (**Right**) Initial velocities are sampled from a $\mathcal{N}(\mathbf{0}, 1 \cdot \mathbf{I})$, i.e. $\sigma_{\boldsymbol{v}_0}^2 = 1.0$. We report the mean and standard error from the mean of the validation Match Rate over 6 seeds. We note that regarding the initial distribution variance, there is a benefit given by enforcing a zero net translation velocity field.

**Score parameterization** We compare the simplified parameterization of the score formulated in Eq. (19) compared to the direct parameterization of Eq. (18). We present in Fig. 1, the validation match-rate during training on the MP-20 dataset. From Fig. 1 *(Center)*, we note that the simplified parameterization leads to faster convergence and better performance. The direct parameterization slowly makes the gap smaller if trained for long enough. We also compare fully trained models on MP-20 and MPTS-52 in terms of MR and RMSE on the test set. We note in Table 3, that no matter the sampling scheme, the simplified parameterization always outperforms the direct one, in addition to being much faster to converge.

Table 3: Impact of the score parameterization on the CSP task performance, on the realistic MP-20 and MPTS-52. The simplified parameterization introduced in Eq. (19) improves significantly upon the direct parameterization from Eq. (18).

| sampler | MP-20 | | MPTS-52 | |
|---|---|---|---|---|
| | MR [%] ↑ | RMSE ↓ | MR [%] ↑ | RMSE ↓ |
| | DIRECT PARAMETERIZATION | | | |
| EM | $56.25_{\pm.4}$ | $0.1071_{\pm.001}$ | $11.55_{\pm.2}$ | $0.2535_{\pm.004}$ |
| PC | $63.29_{\pm.2}$ | $0.0591_{\pm.002}$ | $15.85_{\pm.2}$ | $0.1666_{\pm.004}$ |
| | SIMPLIFIED PARAMETERIZATION | | | |
| EM | $61.72_{\pm.2}$ | $0.0686_{\pm.001}$ | $17.71_{\pm.3}$ | $0.2023_{\pm.005}$ |
| PC | $65.37_{\pm.1}$ | $0.0455_{\pm.001}$ | $21.46_{\pm.2}$ | $0.1339_{\pm.002}$ |

## F. Additional CSP results

Previous work also evaluated their approach on the CARBON-24 (Pickard, 2020) dataset, which consists of carbon-only materials with 6 to 24 atoms per unit cell. Since all structures are composed solely of carbon atoms, computing the MR@1 on the test set is inherently ill-defined—*i.e.* multiple distinct structures may satisfy a specific formula.

For completeness, we report results in Table 4, where KLDM performs comparably to existing methods on the @1 task experiment, but outperforms them on @20. Similar to observations on the PEROV-5 dataset, we find that the PC sampler does not improve sample quality over the standard EM sampler in this setting.

Table 4: Crystal Structure Prediction (CSP) task results on the CARBON-24 dataset. Baseline results are extracted from the respective papers. @ indicates the number of samples considered to evaluate the metrics, *e.g.* @20 indicates the best of 20. Error bars for @1 represent the standard deviation over the mean at sampling time across 20 different seeds. For this dataset, the CSP@1 task is ill-defined due to its one-to-many nature, and we only report the obtained values for completeness.

| | CARBON-24 | |
|---|---|---|
| MODEL | MR [%] ↑ | RMSE ↓ |
| METRICS @ 1 | | |
| CDVAE | 17.09 | 0.2969 |
| DIFFCSP (PC) | 17.54 | **0.2759** |
| EQUICSP (PC) | — | — |
| FLOWMM | **23.47** | 0.4122 |
| **KLDM-$\varepsilon$** (EM) | $18.04_{\pm.8}$ | $0.3188_{\pm.008}$ |
| **KLDM-$\varepsilon$** (PC) | $17.26_{\pm.7}$ | $\underline{0.2827}_{\pm.006}$ |
| **KLDM-$x_0$** (EM) | $\underline{18.58}_{\pm.9}$ | $0.3278_{\pm0.008}$ |
| **KLDM-$x_0$** (PC) | $17.69_{\pm.8}$ | $0.2915_{\pm.007}$ |
| METRICS @ 20 | | |
| CDVAE | 88.37 | 0.2286 |
| DIFFCSP (PC) | 88.47 | 0.2192 |
| FLOWMM | 84.15 | 0.3301 |
| **KLDM-$\varepsilon$** (EM) | **90.19** | 0.2154 |
| **KLDM-$\varepsilon$** (PC) | 85.86 | **0.1988** |
| **KLDM-$x_0$** (EM) | $\underline{89.66}$ | 0.2198 |
| **KLDM-$x_0$** (PC) | 84.58 | $\underline{0.2011}$ |

## G. Additional DNG results

In this section, we present additional results regarding the DNG task. These results were computed using the KLDM-$\varepsilon$ model trained on the non-standardized values of the lattice parameters and implemented using the $\varepsilon$-parameterization for their score. We evaluate samples in terms of validity, coverage, and property statistics. A sample is deemed structurally valid if all pairwise distances are above 0.5Å, while SMACT (Davies et al., 2019) is used to determine compositional validity, by checking the overall charge neutrality. The coverage metrics are computed using fingerprints: CrystalNN structural fingerprints (Zimmermann & Jain, 2020) and Magpie compositional fingerprints (Ward et al., 2016). COV-R (recall) and COV-P (precision) are obtained by comparing the distances between generated and test fingerprints. Finally, the property statistics are obtained by comparing distributions of properties, computed on the generated samples and test set respectively: atomic densities $d_\rho$ and number of unique elements $d_{elem}$.

We provide results in Table 5, where it can be observed that, unlike for the CSP task, the difference in performance between the compared methods is less pronounced. Nevertheless, KLDM remains competitive with the other models.

Table 5: Results for the De-novo Generation (DNG) task. Baseline results are extracted from the respective papers.

| | VALIDITY [%] ↑ | | COVERAGE [%] ↑ | | PROPERTY ↓ | |
|---|---|---|---|---|---|---|
| | STRUC. | COMP. | COV-R | COV-P | $d_\rho$ | $d_{\text{elem}}$ |
| | | | PEROV-5 | | | |
| CDVAE | 100.0 | 98.59 | 99.45 | 98.46 | 0.1258 | 0.0628 |
| DIFFCSP | 100.0 | 98.85 | 99.74 | 98.27 | 0.1110 | 0.0128 |
| EQUICSP | 100.0 | 98.60 | 99.60 | 98.76 | 0.1110 | 0.0503 |
| **KLDM-**$\varepsilon$ | 99.97 | 98.83 | 99.60 | 98.61 | 0.2970 | 0.0478 |
| | | | CARBON-24 | | | |
| CDVAE | 100.0 | – | 99.80 | 83.08 | 0.1407 | – |
| DIFFCSP | 100.0 | – | 99.90 | 97.27 | 0.0805 | – |
| EQUICSP | 100.0 | – | 99.75 | 97.12 | 0.0734 | – |
| **KLDM-**$\varepsilon$ | 100.0 | – | 99.90 | 98.86 | 0.0658 | – |
| | | | MP-20 | | | |
| CDVAE | 100.0 | 86.70 | 99.15 | 99.49 | 0.6875 | 1.432 |
| DIFFCSP | 100.0 | 83.25 | 99.71 | 99.76 | 0.3502 | 0.3398 |
| EQUICSP | 99.97 | 82.20 | 99.65 | 99.68 | 0.1300 | 0.3978 |
| FLOWMM | 96.85 | 83.19 | 99.49 | 99.58 | 0.2390 | 0.0830 |
| **KLDM-**$\varepsilon$ | 99.88 | 84.86 | 98.94 | 99.50 | 0.4658 | 0.1280 |

# H. Algorithms

This section presents all the algorithms at the core of our method. We show how one can sample the noisy inputs and the training targets in Algorithm 1, we then present the training loop used to train the parameters of the score network $s_\theta(\boldsymbol{f}_t, \boldsymbol{v}_t, \boldsymbol{l}_t, \boldsymbol{a}_t)$ in Algorithm 2. We then focus on sampling, presenting both a sampling scheme that uses an exponential integrator for simulating the reverse SDE of the dynamics of the velocities $\boldsymbol{v}_t$ and fractional coordinates $\boldsymbol{f}_t$ as proposed by Zhu et al. (2024) while using a classic Euler–Maruyama step for lattice parameters $\boldsymbol{l}_t$ and atom type $\boldsymbol{a}_t$ in Algorithm 3. In algorithm Algorithm 4, instead, we present how we sample from our model using the predictor-corrector steps as proposed in (Song et al., 2021).

---

**Algorithm 1** `training_targets`$(\boldsymbol{f}, \boldsymbol{v}, \boldsymbol{l}, \boldsymbol{a}, t)$: Routine for sampling $\boldsymbol{f}_t, \boldsymbol{v}_t, \boldsymbol{l}_t, \boldsymbol{a}_t$ from the transition kernels and the corresponding target scores

---

**Require:** task (either CSP or DNG), timestep $t$, scheduler $\alpha(t)$ and $\sigma(t)$ for $\boldsymbol{l}$ and $\boldsymbol{a}$, scheduler $\alpha_{\boldsymbol{v}}(t)$ and $\sigma_{\boldsymbol{v}}(t)$ for $\boldsymbol{v}$, scheduler $\mu_{\boldsymbol{r}_t}(t, \boldsymbol{v}_0, \boldsymbol{v}_t)$ and $\sigma_{\boldsymbol{r}_t}(t, \boldsymbol{v}_0, \boldsymbol{v}_t)$ for $\boldsymbol{r}$. Initial sample $\boldsymbol{x}_0 = (\boldsymbol{f}_0, \boldsymbol{l}_0, \boldsymbol{a}_0)$ and initial velocities $\boldsymbol{v}_0$. In our experiments we considered $\boldsymbol{v}_0 = \boldsymbol{0}$ (i.e. initial velocities are 0).
    ## Sampling $\boldsymbol{v}_t$ and $\boldsymbol{f}_t$
    Sample $\epsilon_{\boldsymbol{v}} \sim \mathcal{N}_{\boldsymbol{v}}(\boldsymbol{0}, \boldsymbol{I}), \epsilon_{\boldsymbol{r}_t} \sim \mathcal{N}_{\boldsymbol{v}}(\boldsymbol{0}, \boldsymbol{I})$          $\triangleright \mathcal{N}_{\boldsymbol{v}}$ is a normal distribution such that $\sum_i \boldsymbol{v}_i = \boldsymbol{0}$.
    $\boldsymbol{v}_t = \alpha_{\boldsymbol{v}}(t)\boldsymbol{v}_0 + \sigma_{\boldsymbol{v}}(t) \cdot \epsilon_{\boldsymbol{v}}$
    **if** initial velocities are zero **then**
        $\text{target}_{\boldsymbol{v}} = -\boldsymbol{v}_t / \sigma_{\boldsymbol{v}}^2(t)$
    **else**
        $\text{target}_{\boldsymbol{v}} = -\epsilon_{\boldsymbol{v}} / \sigma_{\boldsymbol{v}}(t)$          $\triangleright$ See Eq. (26).
    **end if**
    $\boldsymbol{r}_t = w(\mu_{\boldsymbol{r}_t}(t, \boldsymbol{v}_0, \boldsymbol{v}_t) + \sigma_{\boldsymbol{r}_t}(t, \boldsymbol{v}_0, \boldsymbol{v}_t) \cdot \epsilon_{\boldsymbol{r}_t})$          $\triangleright w$ indicates the wrap function.
    $\boldsymbol{f}_t = w(\boldsymbol{f}_0 + \boldsymbol{r}_t)$
    $\boldsymbol{f}_t = \text{center}(\boldsymbol{f}_t)$          $\triangleright \text{center}(\cdot)$ keeps the center of gravity fixed.
    $\text{target}_{\boldsymbol{s}} = (1 - \exp(-t))/(1 + \exp(-t)) \cdot \nabla_{\boldsymbol{r}(\boldsymbol{v})} \mathcal{N}_w$          $\triangleright$ Equivalent computation of Eq. (26).
    $\text{target}_{\boldsymbol{v}} = \text{target}_{\boldsymbol{v}} + \text{target}_{\boldsymbol{s}}$
    ## Sampling $\boldsymbol{l}_t$
    Sample $\epsilon_{\boldsymbol{l}} \sim \mathcal{N}(\boldsymbol{0}, \boldsymbol{I})$
    $\boldsymbol{l}_t = \alpha(t)\boldsymbol{l}_0 + \sigma(t) \cdot \epsilon_{\boldsymbol{l}}$
    $\text{target}_{\boldsymbol{l}} = \epsilon_{\boldsymbol{l}}$
    **if** task is DNG **then**
        ## Sampling $\boldsymbol{a}_t$
        Sample $\epsilon_{\boldsymbol{a}} \sim \mathcal{N}(\boldsymbol{0}, \boldsymbol{I})$
        $\boldsymbol{a}_t = \alpha(t)\boldsymbol{a}_0 + \sigma(t) \cdot \epsilon_{\boldsymbol{a}}$
        $\text{target}_{\boldsymbol{a}} = \epsilon_{\boldsymbol{a}}$
        **return** $(\boldsymbol{v}_t, \boldsymbol{f}_t, \boldsymbol{l}_t, \boldsymbol{a}_t), (\text{target}_{\boldsymbol{v}}, \text{target}_{\boldsymbol{l}}, \text{target}_{\boldsymbol{a}})$          $\triangleright$ Return both noisy samples and training targets.
    **else**
        **return** $(\boldsymbol{v}_t, \boldsymbol{f}_t, \boldsymbol{l}_t), (\text{target}_{\boldsymbol{v}}, \text{target}_{\boldsymbol{l}})$
    **end if**

---

---

**Algorithm 2** Training algorithm

---

**Require:** score network $s_\theta(t, \boldsymbol{f}_t, \boldsymbol{v}_t, \boldsymbol{l}_t, \boldsymbol{a}_t)$, fractional coordinates transition kernel $p_{t|0}(\boldsymbol{f}_t, \boldsymbol{v}_t | \boldsymbol{f}_0, \boldsymbol{v}_0)$, lattice parameters transition kernel $p_{t|0}(\boldsymbol{l}_t | \boldsymbol{l}_0)$, empirical distribution $q(\boldsymbol{x}) = \frac{1}{N} \sum_{i=1}^{N} \delta_{\boldsymbol{x}_i}$ and distribution over initial velocities $p_0(\boldsymbol{v})$. In our experiments we considered $p(\boldsymbol{v}_0) = \delta(\boldsymbol{v}_0)$ (i.e. initial velocities are 0). Losses weights $\lambda_{\boldsymbol{v}}, \lambda_{\boldsymbol{l}}, \lambda_{\boldsymbol{a}}$. We always used $\lambda_{\boldsymbol{v}} = 1$ and $\lambda_{\boldsymbol{l}} = 1$. For DNG task we require also an atom type transition kernel $p_{t|0}(\boldsymbol{a}_t | \boldsymbol{a}_0)$.

    **for** training iterations **do**

        $\boldsymbol{x}_0 = \{(\boldsymbol{f}_0, \boldsymbol{l}_0, \boldsymbol{a}_0)\}_{i=1}^{B} \sim q(\boldsymbol{x}), t \sim \mathcal{U}(t), \boldsymbol{v}_0 \sim p_0(\boldsymbol{v})$             $\triangleright$ $B$ indicates the batch size.

        **if** task is DNG **then**

            $(\boldsymbol{v}_t, \boldsymbol{f}_t, \boldsymbol{l}_t, \boldsymbol{a}_t), (\text{target}_{\boldsymbol{v}}, \text{target}_{\boldsymbol{l}}, \text{target}_{\boldsymbol{a}}) = \texttt{training\_targets}(\boldsymbol{f}, \boldsymbol{v}, \boldsymbol{l}, \boldsymbol{a}, t)$

            $\text{out}_{\boldsymbol{v}}, \text{out}_{\boldsymbol{l}}, \text{out}_{\boldsymbol{a}} = s_\theta(t, \boldsymbol{f}_t, \boldsymbol{v}_t, \boldsymbol{l}_t, \boldsymbol{a}_t)$      $\triangleright$ The network takes $t, \boldsymbol{f}_t, \boldsymbol{v}_t, \boldsymbol{l}_t, \boldsymbol{a}_t$ as input and output all the scores.

            $\mathcal{L}_{\boldsymbol{a}} = \|\text{out}_{\boldsymbol{a}} - \text{target}_{\boldsymbol{a}}\|_2^2$

        **else**

            $(\boldsymbol{v}_t, \boldsymbol{f}_t, \boldsymbol{l}_t), (\text{target}_{\boldsymbol{v}}, \text{target}_{\boldsymbol{l}}) = \texttt{training\_targets}(\boldsymbol{f}, \boldsymbol{v}, \boldsymbol{l}, \text{None}, t)$

            $\text{out}_{\boldsymbol{v}}, \text{out}_{\boldsymbol{l}} = s_\theta(t, \boldsymbol{f}_t, \boldsymbol{v}_t, \boldsymbol{l}_t, \boldsymbol{a}_t)$

        **end if**

        $\mathcal{L}_{\boldsymbol{l}} = \|\text{out}_{\boldsymbol{l}} - \text{target}_{\boldsymbol{l}}\|_2^2$

        $\text{out}_{\boldsymbol{v}} = (1 - \exp(-t))/(1 + \exp(-t)) \cdot \text{out}_{\boldsymbol{v}} - \boldsymbol{v}_t / \sigma_{\boldsymbol{v}_t}^2$         $\triangleright$ Construct the score, see Eq. (19)

        $\mathcal{L}_{\boldsymbol{v}} = \lambda(t)\|\text{out}_{\boldsymbol{v}} - \text{target}_{\boldsymbol{v}}\|_2^2$             $\triangleright$ $\lambda(t)$ computed as Jiao et al. (2023)

        **if** task is DNG **then**

            $\mathcal{L}_{\text{tot}} = \lambda_{\boldsymbol{v}}\mathcal{L}_{\boldsymbol{v}} + \lambda_{\boldsymbol{l}}\mathcal{L}_{\boldsymbol{l}} + \lambda_{\boldsymbol{a}}\mathcal{L}_{\boldsymbol{a}}$        $\triangleright$ Depending on how we model the atom types, $\lambda_{\boldsymbol{a}}$ has different values

        **else**

            $\mathcal{L}_{\text{tot}} = \lambda_{\boldsymbol{v}}\mathcal{L}_{\boldsymbol{v}} + \lambda_{\boldsymbol{l}}\mathcal{L}_{\boldsymbol{l}}$

        **end if**

        Compute gradients of $\mathcal{L}_{\text{tot}}$ with respect to $\theta$ and perform a gradient step.

    **end for**

---

---

**Algorithm 3** Sampling algorithm

---

**Require:** trained score network $s_\theta(t, \boldsymbol{f}_t, \boldsymbol{v}_t, \boldsymbol{l}_t, \boldsymbol{a}_t)$, $N$ discretization steps, $dt = 1/N$ step-size, prior distributions over velocities $p_T(\boldsymbol{v}) = \mathcal{N}_{\boldsymbol{v}}(\boldsymbol{0}, \boldsymbol{I})$, over fractional coordinates $p_T(\boldsymbol{f}) = \mathcal{U}(\boldsymbol{0}, \boldsymbol{1})$, over lattice parameters $p_T(\boldsymbol{l}) = \mathcal{N}(\boldsymbol{0}, \boldsymbol{I})$, and over atom types $p_T(\boldsymbol{a}) = \mathcal{N}(\boldsymbol{0}, \boldsymbol{I})$. We require also the knowledge of the forward drift $f(t)$ and the diffusion coefficient $g(t)$ of the SDEs describing the evolution of $\boldsymbol{l}$ and $\boldsymbol{a}$.

## Note: in the paper we use 0 as index for samples at $t = 0$. However, here it will be a slightly different notation.

Sample from the prior $\boldsymbol{v}_0 \sim \mathcal{N}_{\boldsymbol{v}}(\boldsymbol{0}, \boldsymbol{I})$, $\boldsymbol{f}_0 \sim \mathcal{U}(\boldsymbol{0}, \boldsymbol{1})$, $\boldsymbol{l}_0 \sim \mathcal{N}(\boldsymbol{0}, \boldsymbol{I})$

Set $\boldsymbol{f}_0 = w(\boldsymbol{f}_0)$         $\triangleright$ $w$ indicates the wrap function.

**if** task is DNG **then**

     Sample $\boldsymbol{a}_0 \sim \mathcal{N}(\boldsymbol{0}, \boldsymbol{I})$

**end if**

**for** $n = 1, \ldots, N$ **do**

     **if** task is DNG **then**

         $\text{out}_{\boldsymbol{v}}^{(n-1)}, \text{out}_{\boldsymbol{l}}^{(n-1)}, \text{out}_{\boldsymbol{a}}^{(n-1)} = s_\theta((1 - (n-1) * dt), \boldsymbol{f}_{n-1}, \boldsymbol{v}_{n-1}, \boldsymbol{l}_{n-1}, \boldsymbol{a}_{n-1})$

     **else**

         $\text{out}_{\boldsymbol{v}}^{(n-1)}, \text{out}_{\boldsymbol{l}}^{(n-1)} = s_\theta((1 - (n-1) * dt), \boldsymbol{f}_{n-1}, \boldsymbol{v}_{n-1}, \boldsymbol{l}_{n-1}, \boldsymbol{a}_{n-1})$

     **end if**

     ## Update step for $\boldsymbol{v}$ and $\boldsymbol{f}$

     $\text{out}_{\boldsymbol{v}} = (1 - \exp(-(1 - (n-1)dt)))/(1 + \exp(-(1 - (n-1)dt))) \cdot \text{out}_{\boldsymbol{v}} - \boldsymbol{v}_t/\sigma_{\boldsymbol{v}_t}^2$     $\triangleright$ Follow Eq. (19)

     Sample $\epsilon_{\boldsymbol{v}} \sim \mathcal{N}_{\boldsymbol{v}}(\boldsymbol{0}, \boldsymbol{I})$       $\triangleright$ $\mathcal{N}_{\boldsymbol{v}}$ is a normal distribution such that $\sum_i \boldsymbol{v}_i = \boldsymbol{0}$.

     $\boldsymbol{v}_n = \exp(dt)\boldsymbol{v}_{n-1} + 2(\exp(2dt) - 1)\text{out}_{\boldsymbol{v}}^{(n-1)} + \sqrt{\exp(2dt) - 1}\epsilon_{\boldsymbol{v}}$       $\triangleright$ Update on $\boldsymbol{v}$

     $\boldsymbol{f}_n = w(\boldsymbol{f}_{n-1} - \boldsymbol{v}_n dt)$       $\triangleright$ Update on $\boldsymbol{f}$

     ## Update step for $\boldsymbol{l}$

     Sample $\epsilon_{\boldsymbol{l}} \sim \mathcal{N}(\boldsymbol{0}, \boldsymbol{I})$

     $\boldsymbol{l}_n = \boldsymbol{l}_{n-1} - (f(t) - g^2(t)s(\text{out}_{\boldsymbol{l}}^{(n-1)}))dt + \sqrt{dt}\epsilon_{\boldsymbol{l}}$       $\triangleright$ EM step for $\boldsymbol{l}$

     **if** task is DNG **then**

         ## Update step for $\boldsymbol{a}$

         Sample $\epsilon_{\boldsymbol{a}} \sim \mathcal{N}(\boldsymbol{0}, \boldsymbol{I})$

         $\boldsymbol{a}_n = \boldsymbol{a}_{n-1} - (f(t) - g^2(t)s(\text{out}_{\boldsymbol{a}}^{(n-1)}))dt + \sqrt{dt}\epsilon_{\boldsymbol{a}}$       $\triangleright$ EM step for $\boldsymbol{a}$

     **end if**

**end for**

**if** task is DNG **then**

     **return** A crystalline material sample $(\boldsymbol{f}_N, \boldsymbol{l}_N, \boldsymbol{a}_N)$

**else**

     **return** A crystalline material sample $(\boldsymbol{f}_N, \boldsymbol{l}_N)$

**end if**

---

**Algorithm 4** Sampling with a single Predictor-Corrector step (PC) similar to Rozet & Louppe (2023, Algorithm 4)

---

**Require:** trained score network $s_\theta(t, \boldsymbol{f}_t, \boldsymbol{v}_t, \boldsymbol{l}_t, \boldsymbol{a}_t)$, $N$ discretization steps, $dt = 1/N$ step-size, prior distributions over velocities $p_T(\boldsymbol{v}) = \mathcal{N}_{\boldsymbol{v}}(\boldsymbol{0}, \boldsymbol{I})$, over fractional coordinates $p_T(\boldsymbol{f}) = \mathcal{U}(\boldsymbol{0}, \boldsymbol{1})$, over lattice parameters $p_T(\boldsymbol{l}) = \mathcal{N}(\boldsymbol{0}, \boldsymbol{I})$, and over atom types $p_T(\boldsymbol{a}) = \mathcal{N}(\boldsymbol{0}, \boldsymbol{I})$. We require the scheduler $\alpha_{\boldsymbol{v}}(t)$ and $\sigma_{\boldsymbol{v}}(t)$ for $\boldsymbol{v}$ and also the knowledge of the forward drift $f(t)$ and the diffusion coefficient $g(t)$ of the SDEs describing the evolution of $\boldsymbol{l}$ and $\boldsymbol{a}$. We also require a hyperparameter $\tau > 0$.

## Note: in the paper we use 0 as index for samples at $t = 0$. However, here it will be a slightly different notation.

Sample from the prior $\boldsymbol{v}_0 \sim \mathcal{N}_{\boldsymbol{v}}(\boldsymbol{0}, \boldsymbol{I})$, $\boldsymbol{f}_0 \sim \mathcal{U}(\boldsymbol{0}, \boldsymbol{1})$, $\boldsymbol{l}_0 \sim \mathcal{N}(\boldsymbol{0}, \boldsymbol{I})$      ▷ First steps are similar to Algorithm 3

Set $\boldsymbol{f}_0 = w(\boldsymbol{f}_0)$      ▷ $w$ indicates the wrap function.

**if** task is DNG **then**
     Sample $\boldsymbol{a}_0 \sim \mathcal{N}(\boldsymbol{0}, \boldsymbol{I})$
**end if**

**for** $n = 1, \dots, N$ **do**
     **if** task is DNG **then**
         $\text{out}_{\boldsymbol{v}}^{(n-1)}, \text{out}_{\boldsymbol{l}}^{(n-1)}, \text{out}_{\boldsymbol{a}}^{(n-1)} = s_\theta((1 - (n-1)*dt), \boldsymbol{f}_{n-1}, \boldsymbol{v}_{n-1}, \boldsymbol{l}_{n-1}, \boldsymbol{a}_{n-1})$
     **else**
         $\text{out}_{\boldsymbol{v}}^{(n-1)}, \text{out}_{\boldsymbol{l}}^{(n-1)} = s_\theta((1 - (n-1)*dt), \boldsymbol{f}_{n-1}, \boldsymbol{v}_{n-1}, \boldsymbol{l}_{n-1}, \boldsymbol{a}_{n-1})$
     **end if**
     ## Update step for $\boldsymbol{v}$ and $\boldsymbol{f}$
     ## Prediction step on velocities $\boldsymbol{v}$ and coordinates $\boldsymbol{f}$
     Compute $r = \alpha_{\boldsymbol{v}}(n)/\alpha_{\boldsymbol{v}}(n-1)$
     Compute $c = (r\sigma_{\boldsymbol{v}}(n-1) - \sigma_{\boldsymbol{v}}(n))\sigma_{\boldsymbol{v}}(n-1)$
     $\boldsymbol{v}_n^{\text{pred}} = r\boldsymbol{v}_{n-1} + c\text{out}_{\boldsymbol{v}}^{(n-1)}$
     $\boldsymbol{f}_n^{\text{pred}} = w(\boldsymbol{f}_{n-1} + \boldsymbol{v}_{n-1}^{\text{pred}}dt)$
     ## Correction step on velocities $\boldsymbol{v}$ and coordinates $\boldsymbol{f}$
     **if** task is DNG **then**
         $\text{out}_{\boldsymbol{v}}, \text{out}_{\boldsymbol{l}}, \text{out}_{\boldsymbol{a}} = s_\theta((1 - n*dt), \boldsymbol{f}_n^{\text{pred}}, \boldsymbol{v}_n^{\text{pred}}, \boldsymbol{l}_{n-1}, \boldsymbol{a}_{n-1})$
     **else**
         $\text{out}_{\boldsymbol{v}}, \text{out}_{\boldsymbol{l}} = s_\theta((1 - n*dt), \boldsymbol{f}_n^{\text{pred}}, \boldsymbol{v}_n^{\text{pred}}, \boldsymbol{l}_{n-1}, \boldsymbol{a}_{n-1})$
     **end if**
     $\text{out}_{\boldsymbol{v}} = (1 - \exp(-(1 - n*dt)))/(1 + \exp(-(1 - n*dt))) \cdot \text{out}_{\boldsymbol{v}} - \boldsymbol{v}_t/\sigma_{\boldsymbol{v}_t}^2$      ▷ Follow Eq. (19)
     Compute $\delta = \tau \frac{\dim(\text{out}_{\boldsymbol{v}})}{\|\text{out}_{\boldsymbol{v}}\|_2^2}$
     Sample $\epsilon_{\boldsymbol{v}} \sim \mathcal{N}_{\boldsymbol{v}}(\boldsymbol{0}, \boldsymbol{I})$      ▷ $\mathcal{N}_{\boldsymbol{v}}$ is a normal distribution such that $\sum_i \boldsymbol{v}_i = \boldsymbol{0}$.
     $\boldsymbol{v}_n = \boldsymbol{v}_n^{\text{pred}} + \delta\text{out}_{\boldsymbol{v}} + \sqrt{2\delta}\epsilon_{\boldsymbol{v}}$      ▷ Update on $\boldsymbol{v}$
     $\boldsymbol{f}_n = w(\boldsymbol{f}_n^{\text{pred}} - \boldsymbol{v}_n dt)$      ▷ Update on $\boldsymbol{f}$
     ## Update step for $\boldsymbol{l}$
     Sample $\epsilon_{\boldsymbol{l}} \sim \mathcal{N}(\boldsymbol{0}, \boldsymbol{I})$
     $\boldsymbol{l}_n = \boldsymbol{l}_{n-1} - (f(t) - g^2(t)s(\text{out}_{\boldsymbol{l}}))dt + \sqrt{dt}\epsilon_{\boldsymbol{l}}$      ▷ EM step for $\boldsymbol{l}$
     **if** task is DNG **then**
         ## Update step for $\boldsymbol{a}$
         Sample $\epsilon_{\boldsymbol{a}} \sim \mathcal{N}(\boldsymbol{0}, \boldsymbol{I})$
         $\boldsymbol{a}_n = \boldsymbol{a}_{n-1} - (f(t) - g^2(t)s(\text{out}_{\boldsymbol{a}}))dt + \sqrt{dt}\epsilon_{\boldsymbol{a}}$      ▷ EM step for $\boldsymbol{a}$
     **end if**
**end for**
**if** task is DNG **then**
     **return** A crystalline material sample $(\boldsymbol{f}_N, \boldsymbol{l}_N, \boldsymbol{a}_N)$
**else**
     **return** A crystalline material sample $(\boldsymbol{f}_N, \boldsymbol{l}_N)$
**end if**

---

# I. Experimental details

## I.1. Hardware

All experiments presented in this paper can be performed on a single GPU. We relied on a GPU cluster with a mix of RTX 3090 and RTX A5000, with 24GB of memory.

## I.2. Architecture

As mentioned in Section 3.2, our score network is parameterized using an architecture whose backbone is similar to that of previous work (Jiao et al., 2023, DIFFCSP) which enforces the periodic translation invariant by featurizing pairwise fractional coordinate differences with periodic functions of different frequencies. There are two main differences compared to the architecture of DIFFCSP: as we are coupling the fractional coordinates with an additional auxiliary variable that represents the velocity, we need the network to also take this velocity variable $v_t$ as additional inputs. The network is then outputting the score for all the diffusion processes involved in the model: the score related to the velocities, the one for the lattice parameters, and the one for the atom types. The additional modification we do is to consider a two-layer network instead of a single layer to predict the score related to the velocities. Regarding the network parameters, we considered 4 message-passing layers for PEROV-5, while we increased them to 6 for the remaining three datasets. In all the experiments, we considered the hidden dimension to be 512, the time embedding to be a 256-dimensional vector and we used `SiLU` activation with layer norm. While the presentation in the paper is done in terms of continuous time diffusion models, the implementation is done in discrete time to guarantee an apples-to-apples comparison with baselines such as DIFFCSP and EQUICSP.

DIFFCSP employs a graph-neural network as a score network that adapts EGNN from Satorras et al. (2021) to fractional coordinates. In the following, we are going to present all the components that form this architecture.

**Lattice parameters pre-processing**    We follow the same pre-processing steps for the lattice parameters (both lengths and angles) used by Lin et al. (2024, EQUICSP). Lengths are usually defined from $[0, +\infty)$ while angles are defined in the $(0, \pi)$ interval. However. The diffusion process defined in Eq. (7) operates in the $(-\infty, +\infty)$ domain, and therefore can result in unreasonable lattice parameters. Therefore, we use a logarithmic transformation for the lengths, mapping them from $(0, +\infty)$ to $(-\infty, +\infty)$. For angles, we map them using the following operation $\tan(\phi - \pi/2)$ from $(0, \pi)$ to $(-\infty, +\infty)$.

**Components of EGNN**    Let consider $\boldsymbol{f}_t, \boldsymbol{v}_t, \boldsymbol{l}_t, \boldsymbol{a}$ being the input of our network. The input features are computed by

$$\boldsymbol{h}_i^{(0)} = \mathrm{NN}(f_{\mathrm{atom}}, f_{\mathrm{pos}(t)}),$$

where $f_{\mathrm{atom}}, f_{\mathrm{pos}}$ are the atomic embedding and sinusoidal positional embedding and NN is an MLP.

Then the input features are processed by a series of $s$ message-passing layers that compute

$$\boldsymbol{m}_{ij}^{(s)} = \varphi_m(\boldsymbol{h}_i^{(s-1)}, \boldsymbol{h}_j^{(s-1)}, \boldsymbol{v}, \boldsymbol{l}, \mathrm{SinusoidalEmbedding}(\boldsymbol{f}_j - \boldsymbol{f}_i))$$

$$\boldsymbol{m}_i^{(s)} = \sum_{j=1}^{N} \boldsymbol{m}_{ij}^{(s)}$$

$$\boldsymbol{h}_i^{(s)} = \boldsymbol{h}_i^{(s-1)} + \varphi_h(\boldsymbol{h}_i^{(s-1)}, \boldsymbol{m}_i^{(s)})$$

where $\boldsymbol{m}_{ij}^{(s)}$ and $\boldsymbol{h}_j^{(s-1)}$ represent the messages at layer $s$ between nodes i and j. $\varphi_m$ and $\varphi_h$ are two MLPs. Compared to the DIFFCSP implementation, we want to highlight that we are also passing the velocity $\boldsymbol{v}$ as input.

The SinusoidalEmbedding is a sinusoidal embedding layer defined as

$$\mathrm{SinusoidalEmbedding}(x) := (\sin(2\pi k x), \cos(2\pi k x))_{k=0,\dots,n_{\mathrm{freq}}}^T,$$

with being a $n_{\mathrm{freq}}$ being an hyper-parameter.

After $S$ steps of message passing, we compute all the different scores by doing the following:

$$\boldsymbol{s}_{\boldsymbol{v}}^{(i)} = \varphi_{\boldsymbol{v}}(\boldsymbol{h}_i^{(S)})$$

$$\boldsymbol{s}_{\boldsymbol{l}} = \varphi_{\boldsymbol{l}}\left(\frac{1}{N}\sum_{i=1}^{N}\boldsymbol{h}_i^{(S)}\right)$$

$$\boldsymbol{s}_{\boldsymbol{a}}^{(i)} = \varphi_{\boldsymbol{a}}(\boldsymbol{h}_i^{(S)})$$

where we want to stress that $\varphi_{\boldsymbol{v}}$ is a 2-layer neural network while $\varphi_{\boldsymbol{l}}$ and $\varphi_{\boldsymbol{a}}$ are single-layer MLPs.

For training, we used the same loss weights that were used by DIFFSCP. We consider $\lambda_{\boldsymbol{v}} = 1$ and $\lambda_{\boldsymbol{l}} = 1$ for the CSP task. For the DNG task, instead, as we consider three different ways for modelling the discrete atom type features, we used different weights depending on the modelling choice. If we use one-hot encoding for the atom types, we still rely on the DIFFSCP weights given by $\lambda_{\boldsymbol{v}} = 1$, $\lambda_{\boldsymbol{l}} = 1$, and $\lambda_{\boldsymbol{a}} = 20$. In the case of analog-bits, we used $\lambda_{\boldsymbol{a}} = 1$, while when using discrete diffusion for the atom types, we scale the losses using $\lambda_{\boldsymbol{a}} = 0.33$. The scaling is needed to make the different loss terms have similar magnitudes.

### I.3. Parameters for the KLDM forward process

We kept the drift coefficient $\gamma(t)$ constant at 1 in all the experiments presented in the paper following Zhu et al. (2024). In their experiments, they also tuned the time horizon $T$ of the process in Eq. (12) depending on the considered task. In our experiments for material generation, we kept the time horizon constant at $T = 2$. In terms of the implementation, as we implemented everything in discrete time, we discretize the time in the interval $[0, 2)$ for the diffusion process. For lattice parameters and atom types, we rely on the standard Euclidean diffusion model where the drift and diffusion coefficients are defined by a linear schedule on the interval $[0, 1)$. We trained all the networks using `AdamW` with the default `PyTorch` parameters, without gradient clipping and by performing early stopping based on metrics computed on a subset of the validation set: match-rate for the CSP task and valid structures for the DNG task.

Table 6: Dataset hyperparameters

| INFO | PEROV-5 | CARBON-24 | MP-20 | MPTS-52 |
|---|---|---|---|---|
| Max Atoms | 5 | 24 | 20 | 52 |
| Total Number of Samples | 18928 | 10153 | 45231 | 40476 |
| Batch Size | 1024 | 256 | 256 | 256 |

### I.4. Major difference between KLDM and competitors

In this section, we compare the key differences between our proposed KLDM and the baseline methods discussed in Section 5, namely DIFFCSP, EQUICSP, and FLOWMM. Among these, we think that DIFFCSP is the closest to our approach. The main differences are that we model fractional coordinates differently and we use a linear noise schedule, as opposed to their cosine noise schedule. In addition to that, DIFFCSP employs a matrix representation for the lattice parameters, while we treat them as a vector of six scalars.

EQUICSP builds upon DIFFCSP with two main modifications: it introduces additional losses to ensure the lattice permutation invariance of the learned distributions, and it defines a different noising mechanism called *Periodic CoM-free Noising* scheme. This scheme ensures that the sampled noise does not induce a translation of the center of mass of $\boldsymbol{f}_0$, thereby preserving the periodic translation invariance in the target score. In contrast, we define the forward process on the coupling given by fractional coordinates and associated velocity variables, and we do not account for lattice permutation invariance, leaving that as a direction for future work.

FLOWMM generalizes Riemannian Flow matching for material generation, which, although closely related to diffusion, is a different modeling approach. In addition to that, it considers an informed prior distribution over the lattice parameters, and in the DNG task, it represents atom types using analog bits (Chen et al., 2023), in contrast to DIFFCSP and EQUICSP, which use a one-hot encoding. We present results by considering the same informed prior and also by using three different representations for the discrete atom type features. Additionally, for DNG, FLOWMM provides extra inputs to the score

network that neither KLDM nor the other baselines use. The additional input represents the cosine of the angles between the Cartesian edges between atoms and three lattice vectors

