# OpenReview forum: "Kinetic Langevin Diffusion for Crystalline Materials Generation"
_ICML.cc/2025/Conference — ICML 2025 poster_

### Official Review · Reviewer_WGaL · 2025-03-14

**Overall Recommendation:** 3

**Summary:**

The paper presents Kinetic Langevin Diffusion for Materials (KLDM), a groundbreaking diffusion model designed for generating crystalline materials. KLDM tackles the challenge of modeling fractional coordinates on a hypertorus by introducing auxiliary Euclidean velocity variables,  eliminating the need for approximations inherent in Riemannian diffusion and ensures consistent training objectives. The model is tested on two key tasks—Crystal Structure Prediction (CSP) and De-novo Generation (DNG)—and achieves competitive results compared to state-of-the-art models, especially on large datasets such as MP-20 and MPTS-52.

**Claims And Evidence:**

The claims in this paper are supported by clear evidence.

**Essential References Not Discussed:**

The references in this paper are sufficient, requiring no further supplementation.

**Experimental Designs Or Analyses:**

This paper conducts experiments on the CSP and DNG benchmarks, comparing with the mainstream models. The results serve as evidence supporting the effectiveness of the method.

**Methods And Evaluation Criteria:**

This article provides a fairly detailed explanation of the methodology, and the evaluation is also quite reasonable.

**Other Comments Or Suggestions:**

See "Questions for Authors".

**Other Strengths And Weaknesses:**

Strengths:
1. This paper tackles the challenge of inconsistent training objectives in crystal generation tasks.
2.  The paper aligns both datasets and metrics with prior models, achieving state-of-the-art (SOTA) results in CSP tasks and comparable results in DNG tasks.

Weaknesses:
1. Technically, the diffusion framework they employ is derived from TDM [C], while the backbone model is based on DiffCSP [A]. Although the paper tackles a key issue and achieves good results, the level of technical innovation appears to be somewhat limited.

**Questions For Authors:**

1. Why can a zero-mean $v$ ensure that $f_0$ and $f_t$ share the same group element $g$? Please provide more intuition and explanation.
2. The core operation of this paper is the introduction of the zero-mean $v$, which ensures the consistency of the target score function $s_\theta$. Why does the paper not provide an ablation study for this operation? I believe such an experiment could highlight the key contribution of the work.

**Relation To Broader Scientific Literature:**

This paper focuses on addressing the issue of inconsistent training objectives in crystal generation tasks, a problem that had not been adequately resolved in previous works such as DiffCSP [A] and EquiCSP [B]. Compared to these models, this paper introduces the Kinetic Langevin Diffusion process, inspired by TDM [C]. By incorporating an  auxiliary velocity $v$, the modeling of fractional coordinates is simplified, eliminating the need to focus on Riemannian manifolds. Compared to TDM [3], this paper extends the method to the crystal generation task, which requires considering additional symmetries.

[A] Jiao, Rui, et al. "Crystal structure prediction by joint equivariant diffusion." NeurIPS 2023.

[B] Lin, Peijia, et al. "Equivariant diffusion for crystal structure prediction." ICML 2024.

[C] Zhu, Yuchen, et al. "Trivialized Momentum Facilitates Diffusion Generative Modeling on Lie Groups." ICLR 2025/

**Theoretical Claims:**

The theoretical claims presented in this paper are well-founded.

---

> ### Author Rebuttal · Authors · 2025-04-01
>
> We thank the reviewer for their positive consideration and suggestions to improve the paper. We address questions and comments below.
>
>
> **De-Novo generation task results** Due to the limited character, we have to refer to the answer provided to reviewer **MrGy** about this topic.
>
>
> **Zero-net translation intuition**
> We agree with the reviewer that the intuition was lacking from the submitted manuscript. Here, we provide a simple example to build intuition. We will include it in the updated version.
>
> Consider a datapoint with a single atom in 1D, i.e. $\boldsymbol{f}_0$ consists of just a single coordinate. In this simple setup, every point $\boldsymbol{f}_t$ can be seen as a periodic translation of any another $\boldsymbol{f}^{'}_t$, hence also of the clean sample $\boldsymbol{f}_0$ itself.
>
> With no constraint on the velocity field, the forward dynamics results in noisy samples $\boldsymbol{f}_t$ corresponding to periodic translations of $\boldsymbol{f}_0$ (i.e. almost surely represented by different group elements) with non-zero target scores *pointing back* to $\boldsymbol{f}_0$. Since all $\boldsymbol{f}_t$ effectively represent the same datapoint, modelling this degree of freedom is unecessary.
>
>
> By constraining the velocity field to be zero-mean, the single velocity has to be zero for the constraint to be satisfied. By simulating the forward dynamics, all noisy samples, $\boldsymbol{f}_t$, are exactly $\boldsymbol{f}_0$ (i.e. they share the same group element) with an associated zero target.
>
>
>
> **Ablation of design choices**
> While the proposed constraint of the velocity field is an important part of the paper, we do not see it as being the main contribution. The core of the paper is instead the extension of the TDM framework to crystalline materials generation. To obtain fast convergence and competitive results, we find that zero initial velocities and the resulting simplified parameterization are key elements (see [Figure](https://anonymous.4open.science/r/rebuttal_icml_kldm-36FF/simplified_vs_direct_parametrization.pdf)). We show that non-zero initial velocities (see [Figure](https://anonymous.4open.science/r/rebuttal_icml_kldm-36FF/init_velocity_when_zero_cog.pdf)) systematically lead to subpar performance.
>
> As suggested by the reviewer, we provide an ablation of the effect of the zero net translation for zero initial velocities (see [Figure](https://anonymous.4open.science/r/rebuttal_icml_kldm-36FF/zero_net_translation.pdf)) and non-zero ones (see [Figure](https://anonymous.4open.science/r/rebuttal_icml_kldm-36FF/v0_not_zero_ablation.pdf)). By removing this unecesseray degree of freedom, we observe a benefit in all cases, in particular with non-zero initial velocities.

---

### Official Review · Reviewer_xaaD · 2025-03-14

**Overall Recommendation:** 3

**Summary:**

This paper proposes a new diffusion model for modeling crystalline materials.

The model is built upon a Kinetic Langevin Diffusion on the fractional coordinates, and standard Euclidean diffusion for the lattice vector and atom types (one-hot embedded).

The core contributions of the paper are:
- proposing to use a velocity noising process in the fractional coordinate diffusion to make the noising process itself invariant to fractional translations.
- proposing a simplified score parameterisation for the combined model.
- application of the model to standard benchmark tasks.

**Claims And Evidence:**

For the most part the claims are well supported.

The claim I have most issue with is the discussion around issue in the subsection *Score parametrization and targets*.

It is not clear to me that the issue is that the conditional target scores can be different for different translations of the same $f_t$. Is this not an expected result of score matching? The point of the denoising score matching loss is to minimise the average square error over the conditional scores to give you the score function. Perhaps I have misunderstood the issue?

The solution proposed still seems interesting to me - but in that it reduces the complexity of the learnt function by quotienting out an additional symmetry of the model, namely by the noising process.

This feature is not ablated in the experimental results, although the simplification in the parameterisation of the score function is, and I think it would be quite important to show that this procedure does indeed help with better model performance.

**Essential References Not Discussed:**

None to my knowladge.

**Experimental Designs Or Analyses:**

Overall the design is sound.

I have one small nitpick and that is that for some of the experiments there are error bars computed, and for others there are not. Could the authors explain why?

Additionally, in the De Novo generation task the majority of the methods appear to be very close together in performance for the majority of metrics. Could the authors comment on which of these metrics is most important, and why there is little gain on this task compared to the tasks presented in table 1.

**Methods And Evaluation Criteria:**

The benchmarks are in line with prior work in the area, and appear to be sufficient.

**Other Comments Or Suggestions:**

None

**Other Strengths And Weaknesses:**

I appreciate the value in the combination of previous ideas presented here, and the innovations in the modelling process regarding the score function parameterisation and noising process. The results in Table 1 tasks suggest that the new model does perform better than competitors in some settings.

There are quite a few typos in the paper:
- 248R differ -> defer.
- 407R does not make sense.
For example.

**Questions For Authors:**

Other than the questions posed in other boxes, could the Authors discuss why they think this method appears to be working better than previous methods? Could they discuss why this does not appear to be the case for the denovo task?

Could the Authors discuss if they see impact for this work and the modeling developments outside the application area of crystal structure generation?

**Relation To Broader Scientific Literature:**

The paper builds upon other work in the crystal structure generation literature, and is compared well to other baseline methods such as CDVAE, DIFFCSP, EQUICSP, FLOWMM. The paper is most related to DIFFSCP, where it replaces the diffusion on the fractional coordinates with the Kinetic Langevin Diffusion.

**Theoretical Claims:**

The pieces of analysis in the paper, such as the loss derivation, are correct. There are no other claims made.

---

> ### Author Rebuttal · Authors · 2025-04-01
>
> We thank the reviewer for their positive consideration and suggestions to improve the paper. Thanks for pointing out some typos, we will correct them in the updates version. We address questions and comments below.
>
>
>
> **Invariant network and equivariant target inconsistency** We agree with the reviewer that in settings where no symmetries are involved, there is no *issue* with the denoising score matching loss. In the present case (i.e. target distribution with translational symmetry), the *problem* stems from the use of a periodic translation invariant score network to match an equivariant target. Considering for example a noisy point-cloud and a periodic translated version thereof, these two datapoints are equivalent from the network's perspective while the target scores are going to be different. Although this is averaged out over the course of the training and does not prevent models from learning a useful score approximation (e.g. DiffCSP and MatterGen), this is undesirable. For an intuition on this, refer to the reply *"Zero-net translation intuition"* given to **Reviewer WGaL** and or an alternative discussion, see also [1].
>
> To further support this, we ablate the effect of the zero net translation in the next paragraph.
>
>
> **Ablation on zero net translation and initial zero velocity**  We investigate the effect of the zero net translation in terms of the match rate on the validation set of MP-20 (see [Figure](https://anonymous.4open.science/r/rebuttal_icml_kldm-36FF/zero_net_translation.pdf)), where we obtain (slightly) better results by enforcing zero net translation.
>
> We also present an analysis about the impact of non-zero initial velocities for different variances (see [Figure](https://anonymous.4open.science/r/rebuttal_icml_kldm-36FF/init_velocity_when_zero_cog.pdf) and [Figure](https://anonymous.4open.science/r/rebuttal_icml_kldm-36FF/v0_not_zero_ablation.pdf)). We observe that the zero net translation get better results no matter the initial distribution, and that by forcing the initial velocity to be zero the model converges faster and get better results in terms of match rate on the validation set.
>
>
>
> **Error bars** We agree with the reviewer that we are not consistent as we present errors bars only for some of the experiments. For the baselines, results are taken from the previous papers. We will add error bars also for the DNG task in the updated version. For CSP@20, this was due to the computational cost, but we can add them in the updated version.
>
>
> **New metrics de novo generation task** Due to the limited character, we have to refer to the answer provided to **Reviewer MrGy** about this topic.
>
>
> **Why does this work?** We hypothesize that the added momentum on the fractional coordinates dynamics is the main driver behind the improved performance over DiffCSP.
> We find that zero initial velocities and velocity fields zero net translation are critical for better results and faster convergence. Exploring different noise schedules for the velocities is an interesting direction for further improving KLDM.
>
>
> **Possible future applications** Our model can also be applied to other tasks that involve the generation of periodic systems. A natural application can be surfaces or other lower dimensional periodic systems, e.g. 2D or 1D materials. The generation of metal-organic frameworks (MOF) is another interesting future application, with the main challenge being the additional modelling of rotational frames.
>
> **References**
>
> [1] Lin, Peijia, et al. "Equivariant diffusion for crystal structure prediction." ICML 2024.

---

### Official Review · Reviewer_MrGy · 2025-03-15

**Overall Recommendation:** 4

**Summary:**

This paper proposes a diffusion model tailored for crystalline material generation. It utilizes the specific manifold structure of the data, and applies the framework of Trivialized Diffusion model, which is a diffusion model that works on Lie groups. This framework avoids doing Riemannian diffusion by taking the tangent space and defining the noising process on the velocity, which lies in an Euclidean space, largely simplifies the computation. It demonstrates empirical performance on structure prediction and de novo generation tasks, with comparable performance with existing methods.
## Update after rebuttal
Thank you for adding these empirical results, comparison and explanations. I have raised my score accordingly.

**Claims And Evidence:**

The claims made in the submission are supported by clear and convincing evidence.

**Essential References Not Discussed:**

N/A

**Experimental Designs Or Analyses:**

The experimental designs are sound. The structure prediction and de novo generation make sense. The ablation study shows the simplified parameterization improves the accuracy of the prediction. The paper also mentions “the simplified parameterization” leads to faster convergence, but I did not see quantitative results supporting this.

**Methods And Evaluation Criteria:**

The methods that specifically design diffusion process for the coordinate parametrization of the crystalline data structures makes sense.

For the structure prediction task, it compares RMSE with ground truth and Match Rate. For RMSE computation, I wonder if it considered the symmetry of the coordinates as described in section 2.1.

The Metric for de novo generation makes sense and aligns with literature.

**Other Comments Or Suggestions:**

For de novo generation, the performance is not as good as existing methods. Maybe a future direction would be adding some guidance of those desirable structures.

**Other Strengths And Weaknesses:**

Strengths: The idea of designing the diffusion process and score-matching objective specific to the crystal problem is novel, and the application of the trivialized diffusion model for this data with the group structure is interesting.

Weaknesses: Analysis and result of complexity, convergence, are missing, which would support the benefit of this approach over existing methods. Especially given the fact that it does not outperform them for de novo generation tasks.

**Questions For Authors:**

Can you provide a complexity analysis and comparison with the existing methods, especially the ones using Riemannian Diffusion models on manifolds? Is the matrix exponential step slow to compute, or are they simplified with the trigonometric functions? Compared to existing methods (DIFFCSP), are there less parameters?

**Relation To Broader Scientific Literature:**

How are the key contributions of the paper related to the broader scientific literature? Be specific in terms of prior related findings/results/ideas/etc.

This paper mainly uses the Trivialized Diffusion Model, which enables simpler training of diffusion model for data with a Lie group structure. It provides an interesting direction of designing the diffusion process specific to the algebraic and geometric structure of crystals. It has application in structure prediction and crystal generation.

**Theoretical Claims:**

I checked the main ideas, the transition kernels and objectives, and they make sense. I did not look into the details of the derivation in the appendix.

---

> ### Author Rebuttal · Authors · 2025-04-01
>
> We thank the reviewer for their positive consideration and suggestions for improving the paper. We address their questions and comments below.
>
> **RMSE computation** Similar to previous work, we compute the RMSE of the generated samples wrt. ground truth using `StructureMatcher` from `pymatgen`, after filtering for structural and compositional validity. The algorithm internally accounts for the symmetries in the data.
>
> **Simplified parameterization** To support this, we provide a plot showing the evolution of the match rate on the validation set of MP-20 (see [Figure](https://anonymous.4open.science/r/rebuttal_icml_kldm-36FF/simplified_vs_direct_parametrization.pdf)), where the *simplified* parameterization is shown to converge significantly faster and to higher values than the *direct* one.
>
> **Other design choices ablation** We note that this simplified parameterization is only possible when $\boldsymbol{v}_0=0$. To further support this design choice, we evaluate the effect of the initial velocity standard deviation on the convergence / performance of the model (see [Figure](https://anonymous.4open.science/r/rebuttal_icml_kldm-36FF/init_velocity_when_zero_cog.pdf)). When $\boldsymbol{v}_0\neq0$, the models do not reach convergence within the allocated budget of $3$k epochs -- as for the direct parameterization in the case $\boldsymbol{v}_0=0$.
>
>
> **Architecture compared to previous models** KLDM and DiffCSP are comparable in terms of the NN architecture. We use the same backbone as that of DiffCSP (and EquiCSP), with the minor difference being that now our score network receives an additional input representing the velocity $\boldsymbol{v}_t$, resulting in a limited increase in learnable parameters.
>
>
> **Matrix exponential computation and difference with other Riemannian score based models (RSBMs)** The main difference with DiffCSP is that our diffusion process is defined on the velocity variables and not directly on the fractional coordinates. Our transition kernel has an additional distribution (wrapped normal + normal), resulting in the modelling  of $(\boldsymbol{v}_t, \boldsymbol{f}_t)$ instead of $\boldsymbol{f}_t$ only. Compared to RSBM (Algorithm 1 in [1]), which DiffCSP builds upon, our process (Eq 12., and Algorithm 3 in the submitted manuscript) has an additional momentum term, resulting in velocities displaying some inertia.
>
> Intuitively, this can be thought of as the difference between gradient descent ($\sim$DiffCSP) and gradient descent with momentum ($\sim$KLDM).
>
> Regarding the expontial map, our implementation follows what we presented in Eq. 15 in the paper and Appendix C.2. In the case of a torus, this is simply equivalent to a translation and wrapping operation.
>
> **De-Novo generation task results** We acknowledge the limitations of the presented metrics, and therefore provide more meaningful discovery-related metrics in this new [table](https://anonymous.4open.science/r/rebuttal_icml_kldm-36FF/dng_additional_results.pdf). Given the timeline and the available resources, the evaluation is performed using a machine-learning interatomic potential, based on the open-source [MatterGen pipeline](https://github.com/microsoft/mattergen).
>
>
> For completeness, we compare $3$ ways of performing diffusion on the discrete atom types: continuous diffusion on one-hot encoded atom types (**C**), continuous diffusion on analog bits (**C-AB**), and discrete diffusion with absorbing state (**D**). Notably, when relying on analog-bits or discrete diffusion to model the atom types, KLDM performs better than DiffCSP in terms of RMSD (lower values means generated structures closer to relaxed ones), energy above the hull (lower values means generated materials closer to stability) and stability, while being slightly subpar on S.U.N..
>
> We however note that that the [compared DiffCSP and MatterGen-MP](https://github.com/microsoft/mattergen/tree/main/benchmark/metrics) were trained on a re-optimized version of MP-20 where some chemical elements have been removed, specifically noble gases, radioactive elements and elements with atomic number greater than 84. Samples with energy above the hull bigger than 0.1 eV / atom have also been filtered out. Our model was trained on the original MP-20.
>
> Regarding Mattergen-MP, we believe that the gap can be explained by different elements: (1) a more expressive denoiser operating in real space, (2) a PC sampler on the lattice parameters, and (3) effect of the pre-processing of MP-20.
>
>
> #### **References**
> [1] De Bortoli, Valentin, et al. "Riemannian score-based generative modelling." NeurIPS 2022

---

### Decision · Program_Chairs · 2025-05-01

**Decision:**

Accept (poster)

**Comment:**

This paper addresses the interesting problem of crystalline material generation using the state of the art tools of diffusion modelling. While the reviewers had some initial doubts, the rebuttal addressed all their concerns. The properly conducted evaluations indeed show the advantage of the presented approach. All reviewers recommend acceptance and the AC agrees. AC kindly asks that the paper reflects all discussions provided in the rebuttal.